# HyPE-GT: where Graph Transformers meet Hyperbolic Positional Encodings

## Abstract

Graph Transformers (GTs) facilitate the comprehension of complex relationships on graph-structured data by leveraging self-attention of the possible pairs of nodes. The structural information or inductive bias of the input graph is provided as positional encodings into the GT. The positional encodings are mostly Euclidean and are not able to capture the complex hierarchical relationships of the corresponding nodes. To address the limitation, we introduce a novel and efficient framework, HyPE, that generates learnable positional encodings in the non-Euclidean hyperbolic space that captures the intricate hierarchical relationships of the underlying graphs. Unlike existing methods, HyPE can generate a set of hyperbolic positional encodings, empowering us to explore diverse options for the optimal selection of PEs for specific downstream tasks. Additionally, we repurpose the generated hyperbolic positional encodings to mitigate the impact of oversmoothing in deep Graph Neural Networks (GNNs). Furthermore, we provide extensive theoretical underpinnings to offer insights into the working mechanism of the HyPE framework. Comprehensive experiments on four molecular benchmarks, including the four large-scale Open Graph Benchmark (OGB) datasets, substantiate the effectiveness of hyperbolic positional encodings in enhancing the performance of Graph Transformers. We also consider Coauthor and Copurchase networks to establish the efficacy of HyPE in controlling oversmoothing in deep GNNs.

## 1 Introduction

Graph Transformers (GTs) are earmarked as one of the milestones for modeling the interactions between the node pairs in the graph. As the existing graph neural network suffers from a few glaring shortcomings like oversmoothing (Li et al., 2018), which occurs due to the recursive neighborhood aggregation, oversquashing (Alon & Yahav, 2020), an information bottleneck caused by the exponential growth of information while increasing the size of the receptive field, and bounded expressive power (Xu et al., 2018a; Morris et al., 2019). Graph Transformers confront the limitations by assuming the entire graph is complete and estimating self-attention for all possible node pairs. Yet, such an approach alleviates the limitations; GTs still lose the structural inductive bias, which causes the loss of positional information of the nodes. As an effective solution, the positional encodings as vectors are integrated with the node features to make the respective nodes topologically aware in the graph. Recently, many efforts were made to generate effective positional encodings for the GTs, like spectral decomposition-based learnable encoding (Kreuzer et al., 2021), structure-aware PEs generated from the rooted subgraph or subtree of the nodes (Chen et al., 2022), encoding the structural dependency (Ying et al., 2021), random walk-based learnable positional encodings (Dwivedi et al., 2021), topological positional encodings (Verma et al., 2025), and many more.

The positional encodings derived from the existing works suffer from several critical limitations. For example, the Spectral Attention Network (SAN) (Kreuzer et al., 2021) generates learnable positional encodings by spectral decomposition of the Laplacian matrix. SAN requires high computational time as well as memory, especially for the generation of edge-feature-based Laplacian positional encodings. Structure-aware Transformer (SAT) estimates pairwise attention scores depending on the respective rooted sub-graphs or sub-trees. SAT requires the extraction of multi-hop subgraphs, increasing the pre-processing time and consuming high

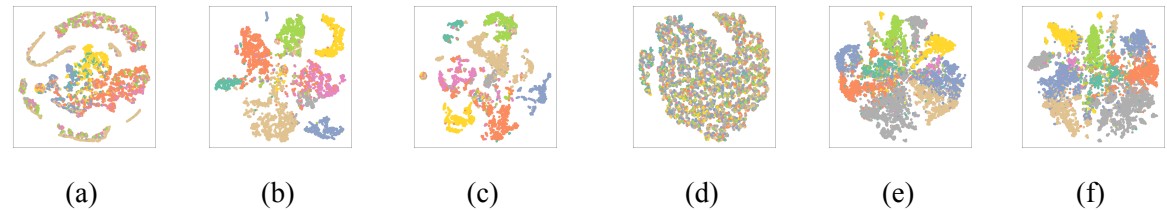

(a)  (b)  (c)  (d)  (e)  (f)

Figure 1: The visualization of node embeddings of Amazon Photo and CoauthorCS generated by 128-layered GCN for the PE category of 4 and 3, respectively. (a) Node embeddings for Amazon Photo without using any positional encodings. (b) Node embeddings of Amazon Photo integrated with the HyPE. (c) Embeddings of hyperbolic PEs from HyPE generated for Amazon Photo. (d) Node embeddings of Coauthor CS without using PEs. (e) Node embeddings of Coauthor CS integrated with the HyPE. (f) Embeddings of hyperbolic PEs from HyPE generated for Coauthor CS.

memory. Even though the depth of the rooted subtree increases, the model may suffer from oversmoothing. Additionally, the proposed methods operate in Euclidean space, which restricts us from exploring the hierarchical relationships of the node pairs in designing positional encodings. This work aims to fill the void by proposing a novel framework named **Hy**perbolic **P**ositional **E**ncodings-based **G**raph **T**ransformer or **HyPE-GT**, which is capable of generating a set of learnable positional encodings in the hyperbolic space. Positional encodings in hyperbolic space can be the appropriate candidates for representing complex and tree-like structures present in the graph. To the best of our knowledge, we are the first to foster hyperbolic geometry in designing positional encodings for the Graph Transformers.

HyPE-GT generates trainable hyperbolic positional encodings, consisting of three pivotal modules (1) the initialization of positional encodings ($PE_{\text{init}}$), (2) the type of hyperbolic manifold (M), where the entire operation will take place, and (3) the employed hyperbolic neural architectures (HNA) which transforming the encodings into the chosen hyperbolic space. Each module can take values from a relevant set of entities and each positional encoding is the result of the unique combination of the entities chosen from the pre-defined sets. The maximum number of positional encodings is given by $|PE_{\text{init}}| \times |M| \times |HNA|$, with the framework offering diverse design choices based on entity selection, generating a wide spectrum of PEs. These hyperbolic positional encodings provide practitioners with versatile options for solving downstream tasks. The learnable hyperbolic PEs can be seamlessly integrated with standard graph transformers, supplying essential positional information. Additionally, we observe that when positional encodings are combined with node features from hidden layers, the effects of oversmoothing can be mitigated in deep GNNs. For each node, the rooted subtree in hyperbolic space generates distinctive embeddings. As the depth increases, the subtree will be well-fitted in that space. Thus incorporation of hyperbolic PEs is instrumental in tackling the oversmoothing issues. Refer to Figure 1 for a vivid illustration of the embeddings of node features and hyperbolic positional encodings.

**Contribution** Our contributions throughout the paper can be summarized in the following way,

- We propose a novel and efficient framework named HyPE-GT that generates a set of learnable positional encodings in the hyperbolic space, a non-Euclidean domain. HyPE-GT is tailored to encode the hierarchical structural patterns into the generated positional encodings and offers better information than Euclidean counterparts. To the best of our knowledge, we are the first to incorporate hyperbolic PEs with the Graph Transformer. The PEs are learned by passing through either hyperbolic neural networks or hyperbolic graph convolutional networks, which produce useful positional and structural encodings.

- The hyperbolic positional encodings are re-purposed to diminish the effect of oversmoothing in deep GNN models. The PEs are incorporated with the transformed features obtained from the hidden layers of GNNs. In this case, the PEs act as distinctive coordinates to avoid potential feature collapse.

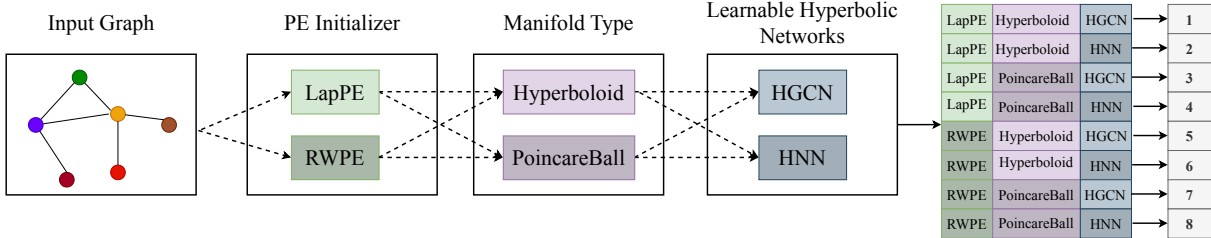

Figure 2: Schematic representation of the process for generating a family of learnable hyperbolic positional encodings (8 different categories) in the HyPE-GT framework. Each category can be generated by following a particular path (shown with arrow-marked dotted lines), which begins from the PE initialization block and ends at the learnable hyperbolic networks block. Each positional encoding is assigned a unique number, shown on the right side of the diagram.

- We present detailed theoretical analyses that offer clarity on the effectiveness of HyPE-GT on graph datasets and the mitigation of oversmoothing in deeper GNNs. The detailed proofs and derivations are discussed in the Appendix.

## 2 Related Works

**Graph Transformers** The attention mechanism on the graph data is primarily introduced with Graph Attention Network (GAT) (Veličković et al., 2017). However, GAT can only learn from the connected neighborhoods via sparse message passing. The limitation was overcome when Dwivedi and Bresson generalized the Transformer architecture for the graphs (Dwivedi & Bresson, 2020) and showed its utility on various categories of tasks. Later, many other significant approaches were taken, like Spectral Attention Network (SAN) (Kreuzer et al., 2021), which relies on learning the eigenvectors of the Laplacian matrix. The learnable eigenvectors act as PE, which is concatenated with the original node features for the Transformer. Structure-aware Transformer (SAT) (Chen et al., 2022) computes the self-attention among node pairs by incorporating structural information of the extracted subgraphs rooted at each node. Another line of work, GraphiT (Mialon et al., 2021), learns relative positional encoding via diffusion kernels, which computes attention between the node pairs. On the other hand, Graphormer (Ying et al., 2021) designs spatial representations like degree encoding, edge encoding, etc., which capture the structural dependency of the graph. The encodings are added as the bias terms in the self-attention matrix. GraphTrans (Wu et al., 2021) proposes a model that first learns from multiple GNN layers stacked together, and then the updated node representations are provided as input to the standard Graph Transformer layer. Recently, another framework, GraphGPS (Rampášek et al., 2022) allows the integration of message-passing models with the module that computes global attention, resulting in an effective and scalable architecture. TokenGT (Kim et al., 2022) considers every node and edge of the graph as independent tokens. The tokens are associated with token embeddings that are the input to the standard Transformer. Edge-augmented Graph Transformer(EGT) (Hussain et al., 2022) introduces edge embeddings for every pair of nodes, which act as the edge gates to control information flow in the Transformer. Graph Transformer Networks (GTN) (Yun et al., 2019) and Heterogeneous Graph Transformer (HGT) (Hu et al., 2020b) are dedicated to heterogeneous graphs, which extract effective meta-paths based on attention. Another work GRIT (Ma et al., 2023) reformulate random walk positional encodings for every pairs of nodes and concatenates them with node features and edge features. Recently, (Verma et al., 2025) designed positional encodings that rely on the topological properties like persistent homology. (Tieu et al., 2025) proposed learnable spatio-temporal positional encodings to improve the performance of link prediction tasks.

**Hyperbolic Graph Neural Networks** Recently, efforts were made to make deep neural networks suitable in the non-Euclidean space like hyperbolic space equipped with negative curvature (Ganea et al., 2018; Khrulkov et al., 2020). Subsequently, the hyperbolic neural networks were extended for the graph-structured

data as Hyperbolic Graph Neural Networks (Liu et al., 2019). On advancement, the hyperbolic graph convolutional networks (Chami et al., 2019) and hyperbolic graph attention networks (Zhang et al., 2021) strengthen the family of the hyperbolic graph neural networks.

## 3 Proposed Method

### 3.1 Preliminaries and Notations

Assume an attributed graph $\mathcal{G} = (\mathcal{V}, \mathcal{E}, X)$, where $\mathcal{V}$ denotes a set of vertices, $\mathcal{E} \subseteq \mathcal{V} \times \mathcal{V}$ denotes set of edges. $X \in \mathbb{R}^{n \times d}$ presents the node attributes where $n$ denotes the number of nodes, and each node is associated with a $d$-dimensional feature. The adjacency matrix $A$ is the symmetric matrix with binary elements denoting the edges or node connections in the graph. $D$ is a diagonal matrix where each element in the diagonal is the degree of the corresponding nodes. Consider $||X||_F$ as the matrix Frobenius norm of $X$.

### 3.2 Transformers on Graphs

Inspired from Vashwani *et al.* (Vaswani et al., 2017), Dwivedi, and Bresson (Dwivedi & Bresson, 2020) extended the philosophy of Transformer for the graph-structured data. Message-passing Graph Neural Networks (MP-GNNs) implement sparse message passing where GTs assume the fully connected graph structure. Unlike MP-GNNs, Graph Transformers are designed to compute the attention coefficient between the pairs of nodes without considering the graph structure. A single Transformer layer consists of a self-attention module followed by a feed-forward network. The feature matrix $X$ is transformed into query $Q$, key $K$, and values $V$ by multiplying with the projection matrices as $Q = XW_Q$, $K = XW_K$, and $V = XW_V$. The self-attention matrix is computed as follows,

$$X_A = \text{softmax}(\frac{QK^T}{\sqrt{d_{out}}})V, \tag{1}$$

where $X_A \in \mathbb{R}^{n \times d_{out}}$ is the self-attention matrix with $d_{out}$ is the output dimension. $W_Q$, $W_K$, and $W_V$ are trainable parameters. To boost the impact of the self-attention module often we employ a multi-head attention (MHA) strategy. The multi-head attention is the result of the concatenation of multiple instances of the Eqn. 1. The multi-head attention can be expressed as:

$$X_A = M \Big\|_{k=1}^{H} \left( \sum_{j \in N(i)} \alpha_{ij}^{(k)} V^{(k)} X_j \right), \tag{2}$$

where $\alpha_{ij}^{(k)}$ is the attention coefficient between node $i$ and $j$ from $k^{th}$ head. $M$ and $V^{(k)}$ are the trainable parameters and $X_j$ is the $j^{th}$ node features. The estimated self-attention matrix is followed by a residual connection and then passes through a feed-forward network (FFN) with the normalization layers as follows,

$$\begin{aligned} X' &= \text{Norm}(X + X_A), \\ X'' &= W_2(\text{ReLU}(W_1 X')), \\ X_o &= \text{Norm}(X' + X''), \end{aligned} \tag{3}$$

where $X_o$ is the final output of the transformer layer and $W_1 \in \mathbb{R}^{d_{out} \times d}$, $W_2 \in \mathbb{R}^{d \times d}$ are trainable parameters. We can either use Batchnorm (Ioffe & Szegedy, 2015) or Layernorm (Ba et al., 2016) for the feature normalization. Each Transformer layer generates node-level representations, which are permutation equivariant. The absence of positional information on the nodes generates similar outputs. Therefore, it is necessary to incorporate appropriate positional encodings to leverage the learning process in the Transformer.

### 3.3 Preliminaries on Hyperbolic Spaces

Hyperbolic geometry deals with the smooth manifold with constant negative curvature. Let us consider the manifold $\mathcal{M} \in \mathbb{R}^d$ embedded in $\mathbb{R}^{d+1}$ with constant curvature $c$. Let us have the following definitions.

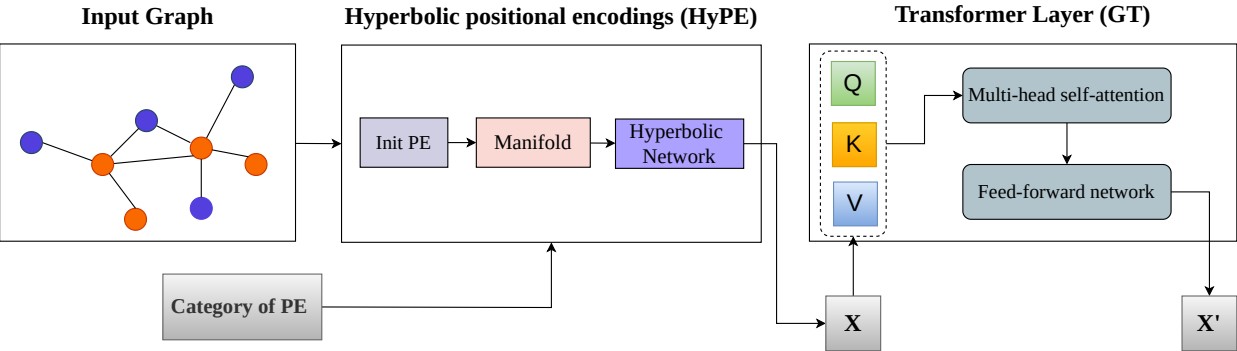

Figure 3: The workflow of HyPE-GT is presented. The framework combines two independent blocks: hyperbolic positional encodings or HyPE and standard Graph Transformer or GT. Depending on one's choice, HyPE will generate a fixed type of PE. The PE is added with the feature matrix $X$ before fetching it to the Transformer layer.

**Tangent space, Logarithmic map, and Exponential map** For every point $x \in \mathcal{M}$, the tangent space $\mathcal{T}_x \mathcal{M}$ is defined as a $d$-dimensional hyperplane approximates the $\mathcal{M}$ around $x$. Define exponential map $\exp_x^c$ as $\exp_x^c : \mathcal{T}_x \mathcal{M} \to \mathcal{M}$ for any point $x$, which transforms any vector in $\mathcal{T}_x \mathcal{M}$ on the $\mathcal{M}$. The logarithmic map $\log_x^c$ is the inverse of the exponential map that is $\log_x : \mathcal{M} \to \mathcal{T}_x \mathcal{M}$, transforms any point on the manifold to the tangent space. The Riemannian metric $g^{\mathcal{M}}$ on the manifold is defined as the inner product on the tangent space like $g^{\mathcal{M}}(.,.) : \mathcal{T}_x \mathcal{M} \times \mathcal{T}_x \mathcal{M} \to \mathbb{R}$. Along with these definitions, in what follows, we will discuss two well-adopted models from hyperbolic geometry.

**Hyperboloid Model** One of the models in the hyperbolic geometry is also commonly known as the Lorentz model. The model is defined as Riemannian manifold $(\mathbb{H}_c^n, g_x^{\mathbb{H}})$ with negative curvature as follows:

$$\mathbb{H}_c^n = \{x \in \mathbb{R}^{n+1} : <x, x>_{\mathbb{L}} = -c, x_0 > 0\} \tag{4}$$

$$g_x^{\mathbb{H}} = diag([-1, 1, \cdots, 1])_n, \tag{5}$$

where $\langle .,. \rangle_{\mathcal{L}}$ denotes Minskowski inner product. The distance between two points $x, y \in \mathbb{H}_c^n$ is defined as

$$d_{\mathcal{L}}^c(x, y) = \sqrt{c} \cosh^{-1}(-\frac{\langle x, y \rangle_{\mathcal{L}}}{c}). \tag{6}$$

For any $v \in \mathbb{H}_c^n$, the exponential and logarithmic maps for the hyperboloid model are as follows:

$$\exp_x^c(v) = \cosh(\frac{||v||_{\mathcal{L}}}{\sqrt{c}})x + \sqrt{c} \sinh(\frac{||v||_{\mathcal{L}}}{\sqrt{c}}) \frac{v}{||v||_{\mathcal{L}}},$$
$$\log_x^c(y) = d_{\mathcal{L}}^c(x, y) \frac{y + \frac{1}{k}\langle x, y \rangle_{\mathcal{L}} x}{||y + \frac{1}{k}\langle x, y \rangle_{\mathcal{L}} x||_{\mathcal{L}}}. \tag{7}$$

**Poincaré Ball Model** Another prominent model that is equipped with negative curvature $c(c < 0)$. The model is defined as the Riemannian manifold $(\mathbb{B}_c^n, g_x^{\mathbb{B}})$ as follows:

$$\mathbb{B}_c^n = \{x \in \mathbb{R}^n : ||x||^2 < -\frac{1}{c}\},$$
$$g_x^{\mathbb{B}} = \frac{2}{(1 + c||x||_2^2)} g^E, \tag{8}$$

where $g^E$ is the Euclidean metric which is $I_n$. The model represents an open ball of radius of $\frac{1}{\sqrt{c}}$. If two points $x, y \in \mathbb{B}_c^n$, then the distance between them is

$$d_{\mathbb{B}}(x, y) = \cosh^{-1}\left(1 + 2\frac{||x - y||^2}{(1 - ||x||^2)(1 - ||y||^2)}\right). \tag{9}$$

For any $v \in \mathbb{B}_c^n$, the exponential and logarithmic maps are defined in the following way:

$$\exp_x^c(v) = x \oplus_c \left( \tanh \left( \sqrt{c} \frac{\lambda_x^c ||v||}{2} \right) \frac{v}{\sqrt{c}||v||} \right),$$

$$\log_x^c(y) = \frac{2\lambda_x^c}{\sqrt{c}} \tanh^{-1}(\sqrt{c}|| - x \oplus y||) \frac{-x \oplus y}{|| - x \oplus y||}, \tag{10}$$

where $\lambda_x^c = \frac{2}{1-c||x||^2}$.

### 3.4 Learnable Hyperbolic Positional Encodings: An Overview

This section presents a brief overview of the HyPE-GT framework for easy apprehension regarding the rest of the work. HyPE-GT consists of three key modules, namely, (1) the initialization of the positional encodings, (2) the type of the manifolds where the transformed PEs will be projected, and the last one is (3) learning the hyperbolic positional encodings by employing either Hyperbolic Neural Network (HNN) or Hyperbolic Graph Convolutional Network (HGCN). As we previously mentioned, the characteristics of the generated positional encodings are nothing but the manifestation of the choice of entities in those three modules. This work considers two entities for each module: LapPE and RWPE for PE initialization, Hyperboloid, and PoincarêBall for manifold type. For learnable networks, we select HNN and HGCN. Refer to Figure 2 for more intricate details of the flow of the framework. Thus, there will be 8 different categories of PEs that can be generated, and each of them is assigned a unique number from $1 - 8$ for ease of reference. The rest of the paper is organized to provide more insights regarding the proposed framework, followed by extensive experimentation.

### 3.5 Initialization of Positional Encodings

The effectiveness of the learnable hyperbolic positional encodings is heavily reliant on the initialization of positional encodings. Despite several existing PEs (Mialon et al., 2021), (Zhou et al., 2020). (Li et al., 2020b), (Ying et al., 2021) etc, we adopt two well-known PE initialization techniques as proposed in (Dwivedi et al., 2021), which are Laplacian Positional Encodings (LapPE) and Random Walk Positional Encoding (RWPE). LapPE is generated by performing eigen-decomposition of the Laplacian matrix of the input graph. For the $i^{th}$ node, the respective LapPE can be presented as:

$$p_i^{\text{LapPE}} = [U_{i1}, U_{i2}, \cdots, U_{ik}] \in \mathbb{R}^k, \tag{11}$$

where $U_i$ denotes the $i^{th}$ eigenvectors of the graph Laplacian and first $k$ elements are chosen resulting in the $k$-dimensional encoding. The $p_i^{\text{LapPE}}$ eigenvector is designated as the initialized position vector for the $i^{th}$ node in the graph. The eigenvectors are assumed to have information on the spectral properties of the input graph. The latter one is the $k$-dimensional RWPE, generated by $k$-steps of the diffusion process with the degree-normalized random walk matrix. For the $i^{th}$ node, RWPE can be represented as follows:

$$p_i^{\text{RWPE}} = [\hat{A}_{ii}, \hat{A}_{ii}^2, \cdots, \hat{A}_{ii}^k] \in \mathbb{R}^k, \tag{12}$$

where $\hat{A} = AD^{-1}$. RWPE captures the structural patterns of the $k$-hop neighborhoods. Finally, the initialization of the PE module has two positional encodings.

### 3.6 Choice of Manifold

The whole mechanism of generating the positional encodings heavily relies on the nature of the manifolds where the initialized positional encodings will be projected. Hence, the choice of a manifold will be one of the critical decisive factors for generating effective PEs as well as the performance enhancement of the Graph Transformers. In HyPE-GT, we will involve two dominant choices named `Hyperboloid` and `PoincarêBall` manifolds to serve the purpose.

Table 1: Performance on three benchmark datasets (Dwivedi et al., 2020). For each category of PE, the results are presented with mean and standard deviations from 10 runs with different random seeds. The category of PE and the optimal number of layers are mentioned, respectively, inside the parentheses. The top three results **first**, second, and third are marked.

| Method | PATTERN | CLUSTER | MNIST | CIFAR10 |
|---|---|---|---|---|
| | Accuracy ↑ | Accuracy ↑ | Accuracy ↑ | Accuracy ↑ |
| GCN (Kipf & Welling, 2016) | $71.892 \pm 0.334$ | $68.498 \pm 0.976$ | $90.705 \pm 0.218$ | $55.710 \pm 0.381$ |
| GIN (Xu et al., 2018a) | $85.387 \pm 0.136$ | $64.716 \pm 1.553$ | $96.485 \pm 0.252$ | $55.255 \pm 1.527$ |
| GAT (Veličković et al., 2017) | $78.271 \pm 0.186$ | $70.587 \pm 0.447$ | $95.535 \pm 0.205$ | $64.223 \pm 0.455$ |
| Gated-GCN (Bresson & Laurent, 2017) | $85.568 \pm 0.088$ | $73.840 \pm 0.326$ | $97.340 \pm 0.143$ | $67.312 \pm 0.311$ |
| PNA (Corso et al., 2020) | – | – | $97.94 \pm 0.12$ | $70.350 \pm 0.630$ |
| DGN (Zhou et al., 2020) | $86.680 \pm 0.034$ | – | – | $72.838 \pm 0.417$ |
| GraphTransformer + LapPE (Dwivedi & Bresson, 2020) | $84.718 \pm 0.068$ | $73.169 \pm 0.622$ | – | – |
| Graphormer (Ying et al., 2021) | – | – | – | – |
| k-subgraph SAT (Chen et al., 2022) | $86.848 \pm 0.037$ | $77.856 \pm 0.104$ | – | – |
| SAN (Kreuzer et al., 2021) | $86.581 \pm 0.037$ | $76.691 \pm 0.65$ | | |
| EGT (Hussain et al., 2022) | $86.821 \pm 0.020$ | $79.232 \pm 0.348$ | $98.173 \pm 0.087$ | $68.702 \pm 0.409$ |
| GraphGPS (Rampášek et al., 2022) | $86.685 \pm 0.059$ | $78.016 \pm 0.180$ | $98.051 \pm 0.126$ | $72.298 \pm 0.356$ |
| Exphormer (Shirzad et al., 2023) | $86.734 \pm 0.008$ | – | $98.414 \pm 0.056$ | $74.754 \pm 0.194$ |
| GRED (Ding et al., 2024) | $86.759 \pm 0.020$ | $78.495 \pm 0.103$ | $98.383 \pm 0.012$ | $76.853 \pm 0.165$ |
| TIGT (Choi et al., 2024) | $86.681 \pm 0.062$ | $78.025 \pm 0.223$ | $98.231 \pm 0.132$ | $73.963 \pm 0.364$ |
| Graph-Mamba (Wang et al., 2024) | $86.710 \pm 0.050$ | $76.800 \pm 0.360$ | $98.420 \pm 0.080$ | $73.700 \pm 0.340$ |
| **HyPE-GT (ours)** | $86.779 \pm 0.038(1, 1)$ | $78.228 \pm 0.126(1, 1)$ | $98.510 \pm 0.007(8, 2)$ | $74.680 \pm 0.009(7, 2)$ |
| **HyPE-GTv2 (ours)** | $86.756 \pm 0.141$ | $78.139 \pm 0.054$ | $98.617 \pm 0.009$ | $74.916 \pm 0.042$ |

## 3.7 Learning through Hyperbolic Neural Architectures

The initialized positional encodings are learned by applying hyperbolic neural network architectures, such as hyperbolic neural networks and hyperbolic graph convolutional networks. Let the initialized positional encodings $p^{\text{init}}$ is transformed to $\hat{p} = W_0 p^{\text{init}}$ into some low dimensional space via the parameterized transformation $W_0$. Then $\hat{p}$ is learnable in the following ways:

**Hyperbolic Neural Network (HNN)** The transformed initial positional encodings are fed into a hyperbolic multi-layered feed-forward neural network. The HNN produces encodings into the hyperbolic space by layer-wise propagation. The transformation can be formulated as follows,

$$p^{\text{HNN}} = \text{HNN}_{\mathcal{M}}^{(L)}(\hat{p} \,|\, \Theta), \tag{13}$$

where $\Theta$ is the trainable parameters of the HNN, $L$ denotes the number of hidden layers, and $\mathcal{M}$ denotes the pre-defined manifold type, which can be either Hyperboloid or PoincarêBall. the learnable positional encodings $p^{\text{HNN}}$ are projected into this manifold. Notably, HNNs only consider the node features as input without considering graph adjacency. Thus, HNNs cannot learn structure-aware embeddings from the input graph. The limitation can be alleviated by using a graph convolutional-based architecture.

**Hyperbolic Graph Convolutional Network (HGCN)** We fed the positional encodings to the hyperbolic graph convolutional networks to extract necessary information from the graph structure. Like the previous one, firstly, the initial positional features are mapped into low-dimensional space, and then positional encodings are learned by the hyperbolic graph convolutional network. The transformation of HGCN can be described as follows:

$$p^{\text{HGCN}} = \text{HGCN}_{\mathcal{M}}^{(L)}(\hat{p} \,|\, \Phi), \tag{14}$$

where $\Phi$ denotes the trainable parameters of the HGCN, $L$ is the number of stacked convolutional layers, and $\mathcal{M}$ can be either Hyperboloid or PoincarêBall model. Note that HGCN takes input from both node features and adjacency and successfully overcomes the shortcomings faced by HNN. The learned PEs will be integrated with the Transformer architecture to provide positional information on the nodes.

## 3.8 Complete Pipeline of HyPE-GT Framework

HyPE-GT comprises two key modules: the first one is HyPE, which generates learnable hyperbolic positional encodings, and the second one is the standard Graph Transformer (GT). HyPE is designed to produce various categories of positional encodings for a single downstream task. The category of PE is decided by

choosing the three components: (1) initialization of positional encodings, (2) the type of the manifold, and (3) the hyperbolic networks. Figure 3 illustrates the complete workflow of generating hyperbolic positional encodings of a given input graph and the integration with the Transformer architecture. For an external input for the category number, we have a pre-defined triplet that produces a fixed category of positional encodings enumerated as $\{1, 2, \cdots, 8\}$. For example, if someone wants to generate the category 4, then the triplet should be like $\{\texttt{LapPE}, \texttt{PoincarêBall}, \texttt{HGCN}\}$. This way, HyPE-GT can produce 8 in diverse categories of positional encodings. However, the generated positional encodings lie in the hyperbolic space, and node features belong to the Euclidean domain. Therefore, the integration of the two should not be straightforward. We devised two different strategies for the addition of hyperbolic PEs with the node features in the following way:

**Incorporating Hyperbolic PEs with GT** The generated PEs from the HyPE pipeline are embedded in the hyperbolic space. Therefore, performing the direct addition between Euclidean node features and hyperbolic positional encodings is not supported. The issue is resolved by transforming the node features into hyperbolic space and is incorporated by performing Möbius addition (Ganea et al., 2018) with the hyperbolic positional encodings. The final output is embedded in the hyperbolic space. If $p_k^{\mathbb{H}}$ is the generated positional encodings in the hyperbolic space with manifold type $\mathbb{H}$ for the $k^{th}$ category and $\hat{x}_v^{\mathbb{E}}$ is the initial node feature in Euclidean space for the $v^{th}$ node, then addition of PEs with node features will be as following:

$$
\begin{aligned}
\hat{x}_v^{\mathbb{H}} &= \exp_0(\tan_{\text{proj}}(\hat{x}_v^{\mathbb{E}})), \\
x_v^{\mathbb{H}} &= \hat{x}_v^{\mathbb{H}} \oplus_c p_k^{\mathbb{H}},
\end{aligned}
\tag{15}
$$

where $\hat{x}_v^{\mathbb{H}}$ and $x_v^{\mathbb{H}}$ are the node features in hyperbolic space before and after addition of the hyperbolic positional encodings respectively, $\oplus_c$ denotes the Möbius addition with space curvature $c$. The function $\tan_{\text{proj}}$ maps the features from the Euclidean domain to the tangent space of the hyperbolic space at the point 0 of the manifold. The updated node features are subsequently reverted into the Euclidean space by applying the log map to provide input to the Transformer.

$$
x_v^{\mathbb{E}} = \log_0(x_v^{\mathbb{H}}),
\tag{16}
$$

where $x_v^{\mathbb{E}}$ is the updated node features in the Euclidean space.

**Utility in Deep GNNs** Oversmoothing (Li et al., 2018) is posed as a dominant challenge of deep GNNs. The node features become indistinguishable due to the recursive neighborhood aggregation during the message propagation in deeper layers. Hence, we re-purpose HyPE-generated hyperbolic positional encodings to couple with the intermediate node features to mitigate the issue. The following operations represent the usage of positional encodings in the output of multi-layered GNNs:

$$
\begin{aligned}
\hat{x}_v^E &= \text{GNN}_L(\hat{x}_v \mid \theta), \\
\hat{x}_v^H &= \exp(\tan_{\text{proj}}(\hat{x}_v^E)), \\
x_v^H &= \hat{x}_v^H \oplus_c p_k^{hyp}, \\
x_v^E &= \log_0(x_v^H),
\end{aligned}
\tag{17}
$$

where $\text{GNN}_L(\hat{x} \mid \theta)$ denotes the output of a $L$-layered GNN architecture with the trainable parameters $\theta$, $x$ is the input node features, $x_o$ is the final output, and $p_k^{hyp}$ is the hyperbolic PE of $k^{th}$ category. $x_v^E$ is the transformed output after adding positional encodings with the node features.

**Preservation of Structural Encodings under Logarithmic Map.** Applying log() map may distort the learned positional encodings in the hyperbolic space. To maintain the structural information, we have theoretically proved that if the norm of the points is bounded by $\frac{1}{\sqrt{c}}$, then log(.) map maintains the isometry with the tangent space. This confirms that the Euclidean representations obtained after applying the logarithmic map aptly preserve the hierarchical information encoded by the hyperbolic positional encodings. We formally represent the theoretical results in *Lemma* 3 of the Appendix.

Table 2: Performance of HyPE-GT on four OGB datasets is presented. The results are the average of 10 different runs, and are reported with the standard deviation. The best category of PE and the number of hyperbolic layers are mentioned in parentheses. The top three results **first**, second, and third are marked.

| Method | ogbg-molhiv | ogbg-ppa | ogbg-molpcba | ogbg-code2 |
| --- | --- | --- | --- | --- |
|  | AUROC ↑ | Accuracy ↑ | Avg. precision ↑ | F1 Score ↑ |
| GCN+virtual node | $0.7599 \pm 0.0119$ | $0.6857 \pm 0.0061$ | $0.2424 \pm 0.0034$ | $0.1595 \pm 0.0018$ |
| GIN+virtual node | $0.7707 \pm 0.0149$ | $0.7037 \pm 0.0107$ | $0.2703 \pm 0.0023$ | $0.1581 \pm 0.0026$ |
| DeeperGCN (Li et al., 2020a) | $0.7858 \pm 0.0117$ | $0.7712 \pm 0.0071$ | $0.2781 \pm 0.0038$ | $0.1570 \pm 0.0032$ |
| GSN (GIN+VN base) (Bouritsas et al., 2022a) | $0.7799 \pm 0.0100$ | - | - | |
| ExpC (Yang et al., 2022) | $0.7799 \pm 0.0082$ | $0.7976 \pm 0.0072$ | $0.2342 \pm 0.0029$ | - |
| GraphTransformer + LapPE (Dwivedi & Bresson, 2020) | $0.7619 \pm 0.0141$ | $0.6864 \pm 0.0047$ | $0.1846 \pm 0.0158$ | $0.1738 \pm 0.0381$ |
| GraphTrans (GCN-virtual) (Wu et al., 2021) | - | - | $0.2761 \pm 0.0029$ | $0.1830 \pm 0.0024$ |
| SAN (Kreuzer et al., 2021) | $0.7785 \pm 0.2470$ | - | $0.2765 \pm 0.0042$ | - |
| GraphGPS (Rampášek et al., 2022) | $0.7880 \pm 0.0101$ | $\mathbf{0.8015 \pm 0.0033}$ | $0.2907 \pm 0.0028$ | $\mathbf{0.1894 \pm 0.0024}$ |
| Specformer (Bo et al., 2023) | $0.7889 \pm 0.0124$ | − | $0.2972 \pm 0.0023$ | − |
| Exphormer (Shirzad et al., 2023) | $0.7834 \pm 0.0044$ | − | $0.2849 \pm 0.0025$ | − |
| GRIT (Ma et al., 2023) | $0.7835 \pm 0.0054$ | − | $0.2362 \pm 0.0020$ | − |
| GECO (Sancak et al., 2025) | $0.7780 \pm 0.0200$ | $0.7982 \pm 0.0042$ | $0.2961 \pm 0.0008$ | $0.1915 \pm 0.0020$ |
| **HyPE-GT (ours)** | $\mathbf{0.7893 \pm 0.0005}(1,2)$ | $0.7981 \pm 0.0043(6,2)$ | $\mathbf{0.2967 \pm 0.0079}(7,2)$ | $0.1855 \pm 0.0054(5,1)$ |
| **HyPE-GTv2 (ours)** | $\mathbf{0.7937 \pm 0.0068}$ | $0.7932 \pm 0.0091$ | $\mathbf{0.2971 \pm 0.0045}$ | $0.1857 \pm 0.0079$ |

## 3.9 Motivation behind choosing the Hyperbolic Space

Hyperbolic space is equipped with constant negative curvature, where the volume of a ball grows in exponential order with respect to the radius. Unlike in the Euclidean domain, where the volume of the ball grows in polynomial order because the space is free of curvature. Therefore, the input graphs can be embedded in the hyperbolic space with lower distortion than in the Euclidean space. In addition, the embeddings in hyperbolic space preserve the neighborhood of every node of the graph. In our framework, HyPE-GT learns positional encodings in the hyperbolic space, which captures the complex patterns of the neighborhoods. Therefore, the positional encodings in the hyperbolic space might be able to represent the topological characteristics corresponding to the nodes in the graph. Refer to Figure 1 for detailed visualizations of our proposed framework's node embeddings in the hyperbolic space. The learned PEs are thereby integrated with the node features as stated in Eqn. 15 16, and 17. Hence, embeddings in the hyperbolic space underscore the effectiveness of the positional encodings, which will be fed to the Graph Transformer.

## 3.10 Theoretical Analysis

In this section, we theoretically analyze the properties of distances in non-Euclidean spaces and their implications for positional encodings in graphs. We establish that distances between points in Poincaré Ball and Hyperboloid spaces are greater than their Euclidean counterparts under specific conditions (Lemma 1 and Lemma 2). Furthermore, for a connected graph, we demonstrate that the distance between the positional encodings of nodes increases when these encodings are transformed via Hyperbolic Neural Networks (HNN) or Hyperbolic Graph Convolutional Networks (HGCN), especially for nodes with higher degrees (Theorem 1 and Theorem 2). All proofs are deferred in the Section Proofs of the Appendix.

### 3.10.1 Distance Properties

**Lemma 1.** *Consider an $n$-dimensional Poincaré Ball $\mathbb{B}^n$ of unit radius and unit curvature. Let us assume that two points $x, y \in \mathbb{B}^n$ and their distance by using Poincaré metric is $d_{\mathbb{B}}(x, y)$. If we apply the Euclidean metric to them, the distance will be $d_E(x, y)$. Then $\forall k \in (0, 1)$, we have $d_{\mathbb{B}}(x, y) \geq 2k d_E(x, y)$ if $d_E(x, y) \in [0, \frac{\sqrt{1-k^2}}{k}]$.*

**Lemma 2.** *Consider an $n$-dimensional Hyperboloid space $\mathbb{H}^n$ of unit radius and unit curvature. Let us assume that two points $x, y \in \mathbb{H}^n$ and their distance by using Hyperboloid distance metric is $d_{\mathbb{H}}(x, y)$. If we apply the Euclidean metric to them, the distance will be $d_E(x, y)$. Then $\forall k \in [1, \infty)$, we have $d_{\mathbb{H}}(x, y) \geq \frac{k}{2} d_E(x, y) \ \forall d_E(x, y) \in [1, \sqrt[4]{\frac{1+k^2}{k^2}}]$.*

**Remark 1.** *Lemmas 1 and 2 suggest that the distance between any two points lying in the non-Euclidean space (here either Poincarê Ball or Hyperboloid space) will be greater than the scaled Euclidean distance estimated between them under certain conditions.*

### 3.10.2 Distinctive Properties of Hyperbolic Positional Encodings via Learning Models

**Theorem 1.** *Consider a pair of nodes $1$ and $2$ of a connected graph $\mathcal{G}$ whose degrees are $d_1$ and $d_2$ respectively. Their initialized positional encodings are $p_1, p_2 \in \mathbb{R}^d$. The Euclidean distance between them is estimated as $d_E(p_1, p_2)$. Suppose, $p_1, p_2$ are to be transformed by either HNN or HGCN with the underlying hyperbolic space as a $n$-dimensional Poincarê Ball $\mathbb{B}^n$ of unit radius and unit curvature, then we have the following:*

1. ***HNN**: If the encodings are transformed by passing through an HNN of parameters $\Theta$. The transformed encodings are respectively $p_1^{hyp}$ and $p_2^{hyp}$ whose distance is $d_{\mathbb{B}}(p_1^{hyp}, p_2^{hyp})$, then $\exists\, \Theta'$ such that $d_{\mathbb{B}}(p_1^{hyp}, p_2^{hyp}) \geq k' ||\Theta'||_F d_E(p_1, p_2)$ for some $k' \in (0, 2)$ and $||\Theta'||_F \leq 1$.*

2. ***HGCN**: If the encodings are transformed by passing through an HGCN of parameters $\Phi$. then $\exists\, \Phi'$ with $||\Phi'||_F \leq 1$ such that $d_{\mathbb{B}}(p_1^{hyp}, p_2^{hyp}) \geq \frac{k'}{d} ||\Phi'||_F d_E(p_1, p_2)$ where $d = \max\{d_1, d_2\}$ for some $k' \in (0, 2)$.*

**Theorem 2.** *Consider a pair of nodes $1$ and $2$ of a connected graph $\mathcal{G}$ whose degrees are $d_1$ and $d_2$ respectively. Their initialized positional encodings are $p_1, p_2 \in \mathbb{R}^d$. The Euclidean distance between them is estimated as $d_E(p_1, p_2)$. Suppose $p_1, p_2$ are to be transformed y either HNN or HGCN with the underlying hyperbolic space as an $n$-dimensional Hyperboloid model $\mathbb{H}^n$ of unit radius and unit curvature, then we have the following:*

1. ***HNN**: If the encodings are transformed by passing through an HNN of parameters $\Theta$. The transformed encodings are respectively $p_1^{hyp}$ and $p_2^{hyp}$ whose distance is $d_{\mathbb{H}}(p_1^{hyp}, p_2^{hyp})$, then $\exists\, \Theta'$ such that $d_{\mathbb{H}}(p_1^{hyp}, p_2^{hyp}) \geq \frac{k'}{2} ||\Theta'||_F d_E(p_1, p_2)$ for some $k' \in [1, \infty)$ and $||\Theta'||_F \leq 1$.*

2. ***HGCN**: If the encodings are transformed by passing through an HGCN of parameters $\Phi$. then there exists a $\Phi'$ with $||\Phi'||_F \leq 1$ such that $d_{\mathbb{B}}(p_1^{hyp}, p_2^{hyp}) \geq \frac{k'}{2d} ||\Phi'||_F d_E(p_1, p_2)$ where $d = \max\{d_1, d_2\}$ for some $k' \in [1, \infty)$.*

**Remark 2.** *Theorems 1 and 2 validate that there exists a parameterized HNN or HGCN architecture that transforms the positional encodings of any pair of nodes in a connected graph such that the distance between them increases under certain conditions. Furthermore, if any node has a higher degree, then it is generally assumed that the node is of higher importance within the graph. Our proposed framework, HyPE, also guarantees the distinctiveness of the PEs when the nodes are of higher degrees.*

## 3.11 An Intuitive Explanation of Theoretical Analyses

In this section, we will intuitively explain the advantages of learning positional encodings in hyperbolic spaces. The theoretical analyses claim the distinctiveness properties of the hyperbolic positional encodings. Consider two nodes $i$ and $j$ in a graph. For a $k$-hop neighborhood, we can construct a rooted subtree for each node. In the hyperbolic space, the rooted subtrees will be better fitted than their Euclidean counterparts. The embeddings of corresponding nodes will capture the essence of tree-like or hierarchical relationships within their own subtrees. The increase in distances between the PE for nodes $i, j$ ensures the distinctive representations compared to their Euclidean counterparts. Thus, PEs are not only enriched with hierarchical information but also leverage unique representations for the tokens (here, nodes) in GTs. Furthermore, the distinctive characteristics of the PEs prevent the potential feature collapse in the multi-layered GNNs.

## 3.12 Complexity Analysis

HyPE-GT consists of three key modules: initialization of PEs, the choice of manifold, and learning through hyperbolic neural networks. We know that $n$ is the number of nodes in the input graph, with the dimension of node features being $d$. The time complexity of the Laplacian positional encodings is $\mathcal{O}(N^3)$. The time complexity of random walk positional encodings will be $\mathcal{O}(dN^3)$.

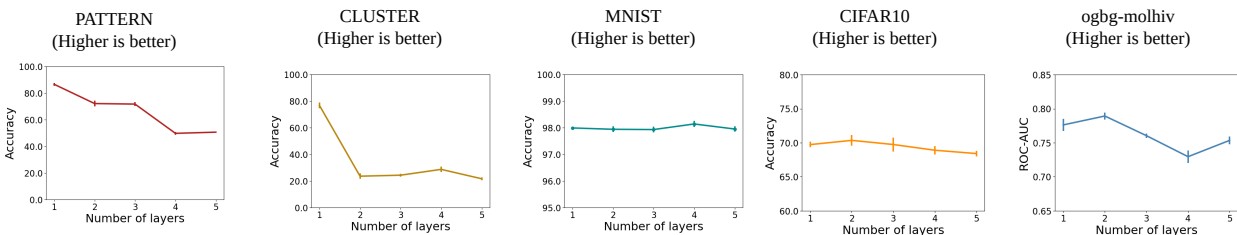

Figure 4: Effect of depth of hyperbolic neural architectures for all five datasets. The number of layers is varied from 1 to 5. For each layer, we averaged the 4 runs on different random seeds.

**Time Complexity of HNN** The HNN performs three consecutive operations: logarithmic map, Möbius matrix-vector multiplication, and exponential map (as defined in Section 3.3). If the logarithmic or exponential map is applied at $x = 0$, then Möbius addition $\oplus_c$ is omitted, thus the complexity is reduced to $\mathcal{O}(d)$. Consider $h, b$ are the $d$-dimensional features and bias vectors in the hyperbolic space, respectively. Therefore, if $W \in \mathbb{R}^{d \times d'}$, then HNN will perform $\exp_0^c(W \log_0^c(h)) \oplus_c b$. The cost of the operation will be $\mathcal{O}(dd' + d + d')$.

**Time Complexity of HGCN** The HGCN will pursue a similar set of operations as described for HNN, except for the neighborhood aggregation. The HGCN will perform $\exp_0^c(\sum_{j \in N(i)} W \log_0^c(h)) \oplus_c b$. The updated operation will cost $\mathcal{O}(d|\mathcal{E}\|)d' + d + d')$.

The computation of Laplacian PEs or random walk PEs is accomplished in the pre-processing stages which is not expected to show impacts on the training time and inference time. During training time, we utilize the estimated positional encodings. The time complexity of the Graph Transformer is $\mathcal{O}(n^2)$. The final complexity of using HNN is $\mathcal{O}(dd')$ and for HGCN is $\mathcal{O}(d|\mathcal{E}|d')$. Incorporating the HyPE framework, the overall time complexity of the graph transformer will cost $\mathcal{O}(n^2 + dd')$ or $\mathcal{O}(n^2 + d|\mathcal{E}|d')$. If $dd' < n^2$, then we have $\mathcal{O}(n^2 + dd') \approx \mathcal{O}(n^2)$. Similarly, if $d|\mathcal{E}|d' < n^2$, then we can write $\mathcal{O}(n^2 + d|\mathcal{E}|d') \approx \mathcal{O}(n^2)$. The tactful selection of the dimension of hyperbolic space will not enhance the overall time complexity of the HyPE-GT compared to the existing standard Graph Transformer.

## 4 Experiments & Results

### 4.1 Datasets

We evaluate HyPE-GT on several benchmark datasets like CLUSTER, PATTERN, MNIST, and CIFAR10 (Dwivedi et al., 2020) for solving node-level and graph-level classification tasks. We further run experiments on large-scale graph datasets like ogbg-molhiv, ogbg-ppa, ogbg-molpcba, and ogbg-code2 from the open graph benchmark (OGB) (Hu et al., 2020a). The performance of deep GNNs is estimated on the co-author and co-purchase datasets (Shchur et al., 2018). The complete details of the datasets can be found in Section 2 of the Supplementary.

### 4.2 Experimental Setup

We consider MNIST and CIFAR-10 to perform classification on the superpixel-based graphs. The classification tasks related to molecular properties are solved on three OGB datasets: ogbg-molhiv, ogbg-ppa, and ogbg-molpcba. On the other hand, the inductive node classification is performed on PATTERN and CLUSTER. Whereas the subtoken prediction task is solved on the ogbg-code2 dataset. The transductive semi-supervised node classification is performed on Amazon Photo, Amazon Computers, Coauthor CS, and Coauthor Physics. The train/valid/test splits for every dataset are provided in Table 3.

We performed experiments on different random seeds and executed 10 different runs. The final results are the average of all runs with standard deviations. The training is stabilized and becomes faster by employing Batchnorm (Ioffe & Szegedy, 2015) added between the layers of the GT. In the case of a multi-layered GNN,

Table 3: The number of instances for each split is provided for 12 datasets used in the experiments.

| Dataset | # train | # valid | # test |
|---|---|---|---|
| PATTERN | 10000 | 2000 | 2000 |
| CLUSTER | 10000 | 1000 | 1000 |
| MNIST | 55000 | 5000 | 10000 |
| CIFAR10 | 45000 | 5000 | 10000 |
| ogbg-molhiv | 3201 | 4113 | 4113 |
| ogbg-ppa | 78200 | 45100 | 34800 |
| ogbg-molpcba | 350343 | 43793 | 43793 |
| ogbg-code2 | 407976 | 22817 | 21948 |
| Amazon Computers | 200 | 300 | 13252 |
| Amazon Photo | 160 | 240 | 7250 |
| Coauthor CS | 600 | 2250 | 15483 |
| Coauthor Physics | 100 | 150 | 34243 |

Layernorm (Ba et al., 2016) is applied for faster convergence of training. The parameters of the models are optimized by Adam. Overfitting is averted by utilizing `ReduceLROnPlateau` during the training process. The data-specific details of the hyperparameters are provided in Section 4 in the Supplementary document. We also perform a comparative study on the number of parameters of HyPE-GT compared to other Graph Transformer-based approaches in Section 6 of the Supplementary document. The Pytorch and DGL-based implementation of the HyPE-GT framework is available at https://anonymous.4open.science/r/HyPE-GT-48D2/README.md.

**Comparison with Baselines.** HyPE-GT can generate 8 different positional encodings depending on the choice of three independent submodules. In the first version, we trained HyPE-GT on eight distinct settings across all datasets and evaluated the test set on each trained model. For each dataset, we reported the best test accuracy among 8 settings. The selection of over 8 choices may incur the higher cost of training time and hyperparameter search. To ensure a fair comparison with competing methods, we now adopt a standardized evaluation protocol: the optimal configuration is first selected based solely on validation performance, and the corresponding model is then evaluated once on the test set. This procedure is replicated across all datasets, and we termed this approach HyPE-GTv2.

### 4.3 Results & Discussion

We will try to resolve the following research questions through empirical evidence.

**RQ1. How do hyperbolic positional encodings improve the performance of the standard graph transformer ?**

HyPE-GT is applied on PATTERN, CLUSTER, MNIST, and CIFAR10 and corresponding results are reported in Table 1. We also applied HyPE-GT on the four OGB datasets whose results are presented in Table 2. Experiments are performed for 8 for different categories of hyperbolic PEs and we reported the best one among them with mean and standard deviation. The corresponding metrics for all datasets are also mentioned.

HyPE-GT attains $3^{rd}$ position on PATTERN with the PE category of 1. PATTERN contains small graphs with the target task of identifying whether a node belongs to a pre-defined pattern. Our framework with a single-layered HGCN successfully captures the pattern in the neighborhoods. Our framework also secures $3^{rd}$ position on CLUSTER with the same category of PE as PATTERN which also contains small-sized graphs. The target task of the CLUSTER is to identify the cluster type to which each node belongs. The task requires information about the local neighborhoods, underscoring that a single-layered HGCN performs optimally. On the other hand, HyPE-GT exhibited superior performance on MNIST for the category 8 outperforming all the contenders. The results suggest the generation of PE by employing HNN entailing the importance of node features rather than the graph structure. Again, HYPE-GT secures $3^{rd}$ position on CIFAR10 for a PE category of 7. CIFAR10 requires higher-order interaction as it is built on RGB images and justifies interactions from the higher-order neighborhoods, underscoring the contribution of a 2-layered HGCN model.

HyPE-GT is further applied on four OGB graphs and the results are presented in Table 2. Our framework outperforms all contenders on ogbg-molhiv for the PE category of 1 with a 2-layered HGCN. ogbg-molhiv contains smaller graphs with chemical hierarchies and sparse edge connectivity. HyPE-GT efficiently captures hierarchies and demonstrates optimal performance. ogbg-molpcba has a structural resemblance to the ogbg-molhiv, with a comparatively larger graph size. HyPE-GT secures $2^{nd}$ position with a PE category of 7 by employing an HGCN architecture. HyPE-GT attains $3^{rd}$ position on ogbg-code2 for the PE category of 5 with a 1-layered HGCN. The dataset contains programming syntax trees where higher-order interactions may not be beneficial. On the other hand, HyPE-GT attains $3^{rd}$ rank on ogbg-ppa but still outperforms DeeperGCN and Transformer+LapPE. The results demonstrate the utility of generating multiple hyperbolic positional encodings, which provides a diverse scope for searching for the optimal positional encodings.

**Performance on Non-Hierarchical Graphs.** In empirical evaluations, we observe that HyPE-GT yields smaller gains on datasets such as PATTERN and CLUSTER, which exhibit limited hierarchical structures. This observation aligns with the geometric intuition that Euclidean spaces are sufficient to represent graphs lacking exponential expansion or multi-scale hierarchy. Interestingly, when curvature $c$ is treated as a learnable parameter, it converges toward zero on these datasets, effectively translating the hyperbolic geometry to a near-Euclidean regime. In this analysis, we obtained the test accuracy for PATTERN and CLUSTER are respectively 83.5149 and 73.9426, which is closer to the performance of Euclidean GT. This phenomenon indicates that HyPE-GT can adapt its representational space to the underlying graph structure—maintaining competitive performance on near-hierarchical graphs while leveraging hyperbolic advantages when hierarchy is present.

Table 4: The performances of GCN, JKNet, and GCNII on co-author and co-purchase networks are presented for different depths of the networks coupled with the HyPE framework. The best results are marked in **green** with the category of PE also mentioned in parentheses, where optimal performance is obtained. (standard deviations are omitted due to space constraints)

| | Method / Layers | 2 | 4 | 8 | 16 | 32 | 64 | 128 |
|---|---|---|---|---|---|---|---|---|
| | GCN | 85.57 | 84.44 | 51.53 | 49.78 | 52.74 | 52.92 | 50.54 |
| | GCN + HyPE | **90.4(7)** | **80.54(8)** | **82.48(2)** | **81.24(3)** | **80.5(3)** | **80.91(3)** | **80.73(3)** |
| Amazon Photo | JKNet | 80.46 | 86.00 | 82.98 | 83.35 | 80.9 | 83.72 | 86.11 |
| | JKNet + HyPE | **89.34(4)** | **90.87(4)** | **90.04(8)** | **89.67(8)** | **89.78(4)** | **89.56(4)** | **90.43(4)** |
| | GCNII | 83.13 | 86.92 | 85.39 | 87.45 | 85.82 | 86.06 | 86.08 |
| | GCNII + HyPE | **91.1(8)** | **92.14(7)** | **92.07(7)** | **91.32(7)** | **91.43(8)** | **91.1(7)** | **91.48(7)** |
| | GCN | 69.94 | 67.55 | 49.0 | 49.38 | 48.6 | 49.55 | 48.66 |
| | GCN + HyPE | **82.61(8)** | **75.78(4)** | **72.86(3)** | **72.6(3)** | **73.97(3)** | **72.91(3)** | **73.75(3)** |
| | JKNet | 64.88 | 74.03 | 53.21 | 54.57 | 58.26 | 55.05 | 67.7 |
| | JKNet + HyPE | **80.03(5)** | **81.13(4)** | **76.86(4)** | **76.49(3)** | **75.8(3)** | **76.12(3)** | **75.63(3)** |
| Amazon Computers | GCNII | 71.93 | 74.58 | 64.04 | 72.59 | 69.54 | 69.99 | 68.68 |
| | GCNII + HyPE | **82.66(8)** | **82.86(7)** | **81.48(7)** | **81.75(4)** | **80.55(7)** | **81.07(4)** | **80.75(7)** |
| | GCN | 89.89 | 83.83 | 16.42 | 15.37 | 12.01 | 21.36 | 11.66 |
| | GCN + HyPE | **92.09(8)** | **88.96(3)** | **82.58(4)** | **82.04(4)** | **82.16(3)** | **82.17(4)** | **81.52(3)** |
| | JKNet | 91.45 | 89.5 | 89.05 | 87.99 | 88.39 | 87.6 | 87.28 |
| | JKNet + HyPE | **92.64(2)** | **92.58(7)** | **92.11(4)** | **92.22(8)** | **92.31(8)** | **92.24(8)** | **92.17(7)** |
| Coauthor CS | GCNII | 90.74 | 90.14 | 88.55 | 92.82 | 93.02 | 93.08 | 93.08 |
| | GCNII + HyPE | **93.19(5)** | **93.01(8)** | **93.5(3)** | **93.65(3)** | **93.58(4)** | **93.68(4)** | **93.58(8)** |
| | GCN | 93.9 | 90.97 | 89.46 | 51.81 | 52.96 | 49.29 | 51.8 |
| | GCN + HyPE | **94.26(4)** | **93.49(3)** | **90.26(4)** | **89.88(2)** | **90.16(4)** | **89.7(2)** | **90.01(2)** |
| | JKNet | 93.56 | 93.31 | 92.77 | 91.99 | 92.29 | 93.4 | 92.09 |
| | JKNet + HyPE | **94.23(7)** | **94.44(8)** | **94.23(7)** | **94.33(7)** | **93.93(5)** | **94.37(8)** | **94.23(8)** |
| Coauthor Physics | GCNII | 94.0 | 93.54 | 93.97 | 94.1 | 94.24 | 94.03 | 94.02 |
| | GCNII + HyPE | **94.37(3)** | **94.45(4)** | **94.6(6)** | **94.62(3)** | **94.53(4)** | **94.43(3)** | **94.76(6)** |

**RQ2. Can hyperbolic positional encodings improve the performance of deep GNN models ?**

Graph Transformer leverages long-range interaction by measuring the attention among the node pairs which tackles the oversmoothing issue to some extent. Yet, we want to explore the effect of incorporating hyperbolic positional encodings directly with the multi-layered graph convolution-based architectures. The generated positional encodings are flexibly integrated with the learned node features from the hidden layers to boost the model performance. We carry out an extensive experiment on co-purchase and co-author datasets Amazon Photo, Amazon Computers, Coauthor CS, and Coauthor Physics aiming to solve the task of semi-supervised mode classification. Our experiment encompasses the three well-adopted base GNN models like GCN (Kipf & Welling, 2016), JKNet (Xu et al., 2018b), and GCNII (Chen et al., 2020). For every dataset, we applied three chosen base models. For each base GNN model, once we run experiments without using any positional encodings and in the second phase HyPE is coupled with the corresponding base model. The process is repeated for every network depth chosen from the set $\{2, 4, 8, 16, 32, 64, 128\}$. The numerical results are reported in Table 4. The performance is measured with the test accuracy which is obtained by taking the mean of the 10 different runs on multiple random seeds. We run experiments for all 8 categories and we only report the optimal one among them. The category of PE is also mentioned along with the highlighted test accuracy for HyPE-GT.

The reported results are evident for the better applicability of the hyperbolic positional encodings integrating with deeper GNN architectures. The base models witnessed an uptick in performance when associated with the HyPE framework compared with the performance of the same without involving the positional encodings at every network depth. The optimal results are obtained for different categories of PEs which also indicates the benefits of generating a diverse set of hyperbolic positional encodings. The node features in the embedding space collapse with the increase of convolutional layers due to the decrease in the inter-cluster distance. The incorporation of positional encodings with the features from the hidden layers prevents the features from collapsing signifying the control of oversmoothing. Notably, the hyperbolic PEs are sufficiently capable of separating similar nodes toward each other, increasing inter-cluster distance. The performance of vanilla GCN is typically hindered by the effect of oversmoothing which is alleviated by employing the HyPE framework. Again the performance improvement on GCNII and JKNet is lesser than GCN due to those models are already designed for controlling oversmoothing but our framework is still able to outperform with a good margin which underlines the efficiency of the proposed framework.

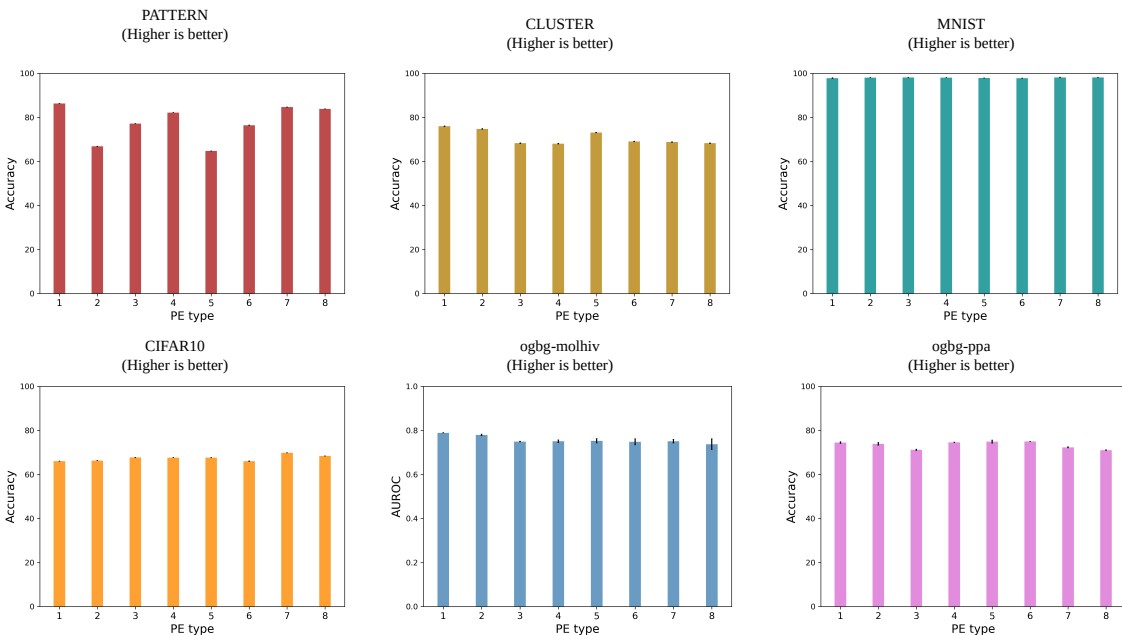

Figure 5: The performance of HyPE-GT on six benchmark datasets is presented. Each possible combination of the key modules in the framework is considered. All eight types of positional encodings are generated for each dataset.

### 4.4 Depth of the Hyperbolic Networks in HyPE-GT

We investigated the effect of the depth of hyperbolic networks (both HNN and HGCN) on the performance of HyPE-GT. We include PATTERN, CLUSTER, MNIST, CIFAR10, and ogbg-molhiv to conduct the experiments. HyPE-GT generates 8 different positional encodings for every dataset and we only run the experiment for the category where the best performance is obtained. For example, MNIST attains the best performance for the PE category of 8. The best category can be found in the results available in Table 1 and 2. We vary the number of layers of the respective hyperbolic networks from 1-5. For each network depth, we run the experiments for 4 times and plot the mean along with standard deviations in Figure 4. The performance metric is also mentioned against each plot.

The performance of HyPE-GT deteriorates on PATTERN and CLUSTER due to the oversmoothing issue in the hyperbolic graph convolutional-based architectures. Yet, these datasets contain graphs with smaller radii and performance depends on local substructures. Thus, increasing depth might not be beneficial for the performance gain. HyPE-GT exhibited stable performance on MNIST as it hinges on the HNN and mostly relies on the rich node features. On the other side, the performance of HyPE-GT on CIFAR10 is optimal when the network depth is 2. The superpixel graphs of CIFAR10 are created from RGB images which requires aggregating features from higher-order neighborhoods. Thus, the degradation of performance is lower compared to PATTERN and CLUSTER. The performance on ogbg-molhiv is optimal when the network depth is 2. The molecular graphs containing local substructures prefer the localized feature aggregation which is further validated by the performance degradation with increased network depth.

## 5 Ablation Study

We conduct a comprehensive ablation study on our proposed framework HyPE-GT to analyze the impact of individual modules within the framework. HyPE-GT comprises three modules as discussed earlier. We explore various options for each of the modules which prompts the generation of 8 different positional encodings. We run the experiments on PATTERN, CLUSTER, MNIST, CIFAR10, ogbg-molhiv, and ogbg-ppa, and the results are presented in Figure 5. The ablation study on ogbg-molpcba and ogbg-code2 can be found in Section 5 of the Supplementary document. The experiments are executed for each category of positional encodings for every dataset. Each result is the average of 10 runs along with the standard deviation with different random seeds. The variation in the corresponding performance metric can be observed across 8 different positional encodings. Notably, optimal performances are obtained for certain combinations of individual modules in the framework, highlighting the interdependence of the modules. The capacity to generate multiple PEs of HyPE broadens the scope of finding the best one for solving target tasks.

### 5.1 Selection Criteria of Positional Encodings

An obvious question will naturally arise regarding the determination of the optimal triplet from the set of positional encodings for solving the downstream tasks. We attempt to resolve the issue by analyzing the experimental results. We devised an intuitive strategy to minimize the search time and avoid the time-consuming effort of searching randomly for the best positional encoding. Our framework achieves optimal results on PATTERN, CLUSTER, CIFAR10, and ogbg-molhiv when the PEs are learned via HGCN rather than HNN. Therefore, HGCN may be an appropriate candidate to initiate the search as it captures the structural patterns of the input graph. Only HyPE-GT on MNIST shows the best performance when HNN is employed; also, the second-best result occurred with HNN. Experiments suggest that Hyperboloid dominates over PoincarêBall in most cases. Furthermore, LapPE and RWPE both work well in the experiments. We still recommend an exhaustive search to find the most effective positional encodings for the downstream tasks.

## 6 Effect of the Readout Methods

The selection of the readout methods is pivotal for performing the graph classification tasks. The fact prompts us to study the effects of three well-adopted readout methods `Mean`, `Max`, and `Sum` when HyPE-

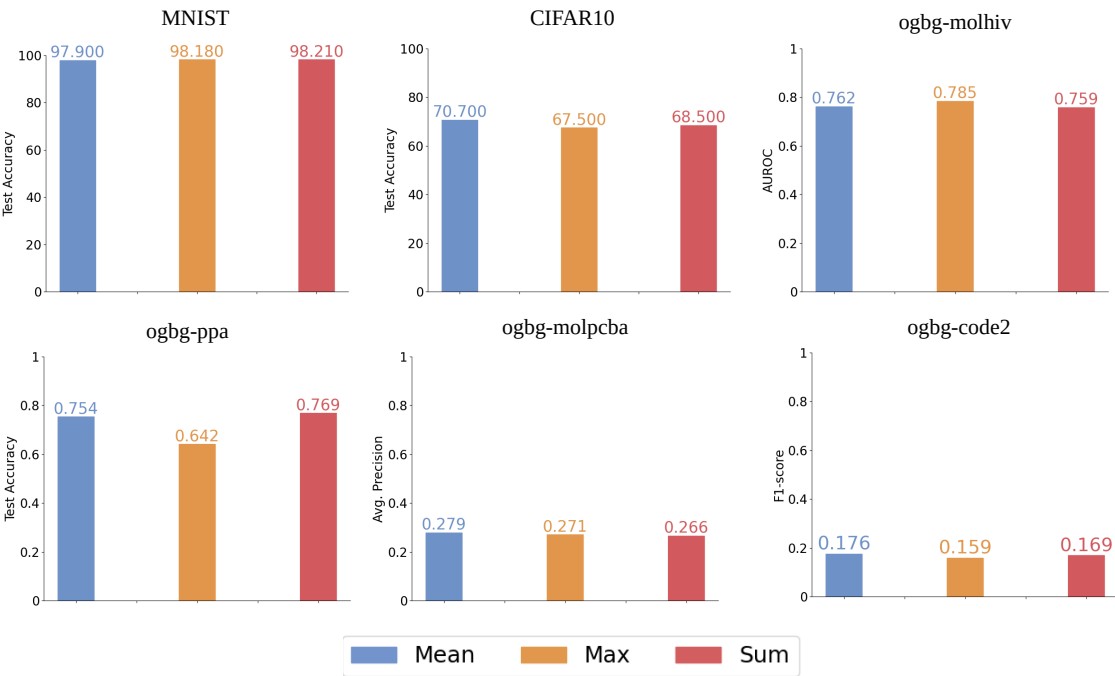

Figure 6: Various readout methods like Mean, Max, and Sum are applied on both 6 graph classification datasets, and their effects are presented. The corresponding metrics are mentioned. The optimal performance can be obtained by tactfully selecting the readout methods which entirely depends on the input graphs.

GT is applied on the MNIST, CIFAR10, ogbg-molhiv, ogbg-moppa, ogbg-molpcba, and ogbg-code2. The performance of HyPE-GT on different readout methods applied on 6 datasets is elucidated in Figure 6. The Sum readout emerged as beneficial for MNIST but the Mean readout is more effective for CIFAR10. Similarly, ogbg-molhiv, ogbg-ppa, ogbg-molpcba, and ogbg-code2 exhibited optimal performances when the readout methods were Max, Sum, and Mean, respectively. The variations among the readout methods across all datasets are relatively lower, which emphasizes the resilience of HyPE-GT toward the choice of readout methods. The experiments also underscore the importance of choosing appropriate readout methods to obtain the best performance on the graph classification tasks.

## 7 Runtime and Memory Profiling

**Experimental Protocol** We conducted a detailed experiment to analyze the effects of hyperbolic operations on the different aspects of Graph Transformer, like (1) average training time, (2) inference time, (3) peak GPU memory consumption, and (4) number of model parameters. We considered the total 8 datasets and ran each experiment once to report the aforementioned parameters. We keep the set of hyperparameters as described in the Appendix to maintain a fair comparison. In this experiment, we only demonstrated the results on the HGCN model projected on the Hyperboloid manifold initialized with Laplacian positional encodings. The space curvature is set to 1.

**Discussions** The comparative results are reported in the Table 5 experimented across 8 datasets. We compared the performances between Euclidean GT and HyPE-GT in four different aspects. The average training time for each dataset is marginally higher for HyPE-GT compared to Euclidean GT. Furthermore, the inference time is not significantly higher for the hyperbolic variant, indicating the utility in real-world datasets. The cost of higher inference time is compensated by the enhanced performance in HyPE-GT. Even the peak memory consumption has minor differences across the datasets, demonstrating the memory

Table 5: A comparative study is performed between Euclidean GT and HyPE-GT for 4 empirical variables. The experiments are executed on 8 graph datasets.

| Dataset | Model | Train Time / Epoch (s) | Inference Time / Batch (s) | Peak GPU Memory (GB) | Params |
|---|---|---|---|---|---|
| MNIST | Euclidean GT | 20.1566 | 2.1964 | 1.2355 | 352670 |
| | HyPE-GT | 26.8915 | 2.6568 | 1.3982 | 395310 |
| CIFAR10 | Euclidean GT | 20.1814 | 2.7403 | 2.0244 | 361142 |
| | HyPE-GT | 26.2809 | 3.2433 | 2.2896 | 395470 |
| PATTERN | Euclidean GT | 54.1691 | 5.9091 | 7.6786 | 944698 |
| | HyPE-GT | 57.4382 | 6.189 | 7.7086 | 983383 |
| CLUSTER | Euclidean GT | 40.8394 | 2.238 | 6.3683 | 965648 |
| | HyPE-GT | 43.2107 | 2.3428 | 6.3988 | 983787 |
| ogbg-molhiv | Euclidean GT | 35.4196 | 2.7596 | 1.6709 | 350917 |
| | HyPE-GT | 43.5797 | 3.0648 | 1.6979 | 389441 |
| ogbg-ppa | Euclidean GT | 73.6192 | 3.7789 | 5.7531 | 9127139 |
| | HyPE-GT | 77.1826 | 4.0127 | 6.8192 | 9396274 |
| ogbg-molpcba | Euclidean GT | 84.1512 | 4.6067 | 2.3595 | 3963250 |
| | HyPE-GT | 102.9816 | 5.2868 | 3.2477 | 4298308 |
| ogbg-code2 | Euclidean GT | 506.0721 | 15.851 | 25.524 | 9373174 |
| | HyPE-GT | 580.7442 | 17.238 | 6.2737 | 11396274 |

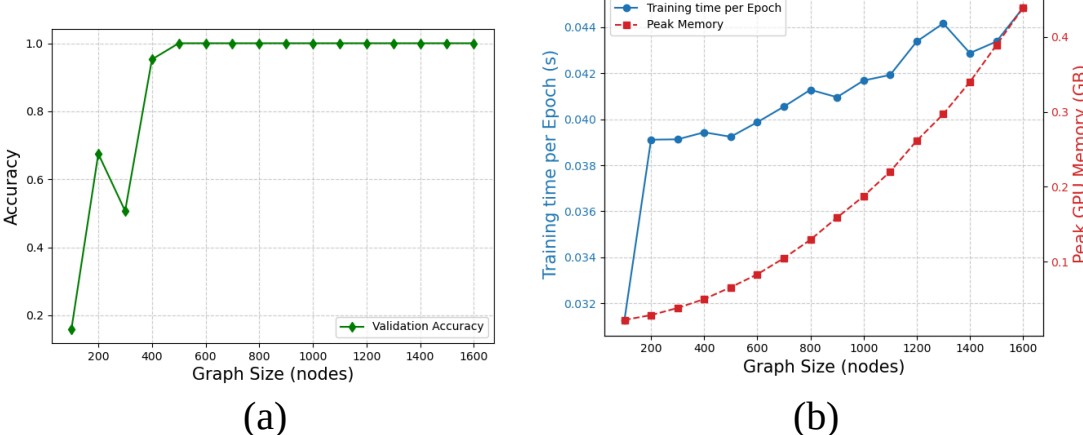

(a)         (b)

Figure 7: HyPE-GT is applied to a set of 16 synthetic graphs with an increasing number of nodes. In (a), the validation accuracy gradually improves with an increasing number of nodes. In (b), the peak memory consumption and training time per epoch follow the quadratic complexity with respect to $n$. The plots follow the theoretically estimated time complexity for HyPE-GT.

efficiency of the proposed HyPE-GT. The parameter complexity in both conditions is highly comparable, which underscores the efficient incorporation of hyperbolic functions in the GT.

# 8 Scalability with Graph Size

**Experimental Protocol** We conducted experiments on a set of synthetic graphs to validate the time complexity of $\mathcal{O}(n^2)$. We generated 16 Erdős-Rény graphs with an increasing node size considered from the set $\{100, 200, 300, \cdots, 1600\}$ and applied HyPE-GT individually on each graph. We set the space curvature to 1. The training continued for 25 epochs for each graph, and we recorded the average training time, peak memory consumption, and validation accuracy. Our downstream task is node classification, where each node is assigned a class label ranging from 0 to 3, depending on the degree of that node. Each node feature is randomly initialized with a 64 dimensional vector.

**Discussions** The experiment is conducted across 16 synthetic graphs. The validation accuracy of HyPE-GT is presented in Figure 7(a). The average training time and memory profiling are presented in Figure 7(b).

With the increasing number of nodes, the validation accuracy monotonically increases and finally reaches 100%. This demonstrates the applicability of our proposed architecture to large-scale graphs. Concurrently, the peak memory consumption with an increasing node size maintains a quadratic complexity bound that validates the asymptotic complexity analysis. Notably, the training time per epoch also maintains a similar trend that aligns with the $\mathcal{O}(n^2)$ time complexity. In essence, the results are evident that the performance of HyPE-GT remains almost unaffected by the hyperbolic functions, even with increasing graph size.

## 9 Analysis on Controlled Hierarchy

**Experimental Protocol** We experimented to investigate the impact of hierarchy on the performance of HyPE-GT on synthetic graphs. We employed three different metrics to estimate the degree of hierarchy, namely (1) tree-based hierarchy, (2) spectral-based hierarchy, and (3) topology-based hierarchy. To generate synthetic graphs with controlled hierarchy, we controlled two parameters, namely (1) depth and (b) branching of the tree. By varying two parameters, we generate a set of 20 with five depths and four different branching factors. The node features are randomly initialized with 32-dimensional vectors, and the size of the positional encodings is set to 8. To initialize the positional encodings, Laplacian eigenvectors are utilized and learned within the HGCN architecture, which projects onto the Hyperboloid manifold. We set the space curvature to 1 in the entire experiment. The hierarchy is controlled by inserting random edges between the different levels in the graphs.

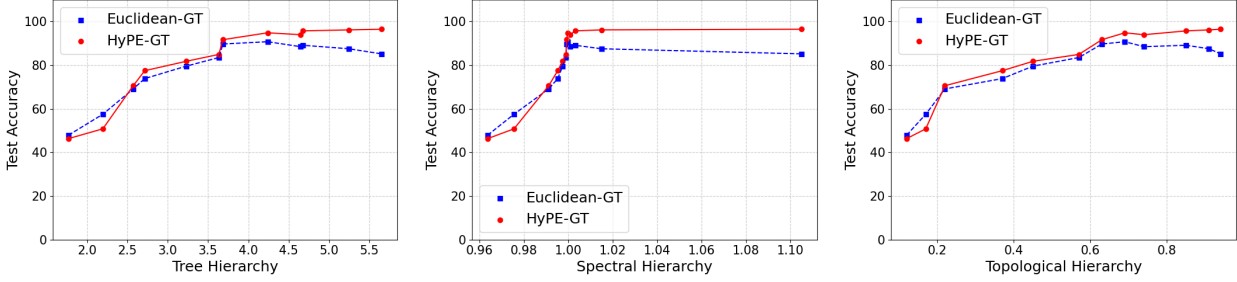

Figure 8: The comparative performance of Euclidean GT and HyPE-GT on hierarchical synthetic graphs is illustrated. With increasing hierarchy values (for all three metrics), the increase in relative performances is observed, ensuring the efficacy of hyperbolic PEs to capture hierarchies.

**Discussions** In Figure 8, the performance trends of Euclidean GT and HyPE-GT remain identical across three metrics. At the initial stage, where the graph hierarchy is lower, Euclidean GT demonstrated efficacy over the HyPE-GT. When the hierarchy increases, the performance of HyPE-GT outperforms its Euclidean variant. The performance trends underline the adeptness of HyPE-GT to capture intricate hierarchical patterns embedded in the underlying graphs. Moreover, the performance of Euclidean GT degrades with an increasing graph hierarchy.

## 10 Statistical Significance Test

**Experimental protocol** We conducted a statistical significance test to analyze the performance of HyPE-GT and other competing methods vividly. The outcome of the test is represented by the Critical Difference (CD) diagram, where the ranks for each method are listed along a horizontal line. The methods with higher ranks are placed on the left side. For implementation, we used the function `critical_difference_diagram` imported from the package `scikit-posthocs`. We set the $\alpha = 0.05$ that decides whether two algorithms are statistically distinct or not.

**Discussions** The critical diagrams for the Tables 1 and 2 are respectively elucidated in the Figures 9(a) and 9(b). The diagrams reveal that our method, HyPE-GT, obtained first rank among the remaining 7 contender algorithms. The corresponding ranks are mentioned for every method in the respective diagrams. The CD

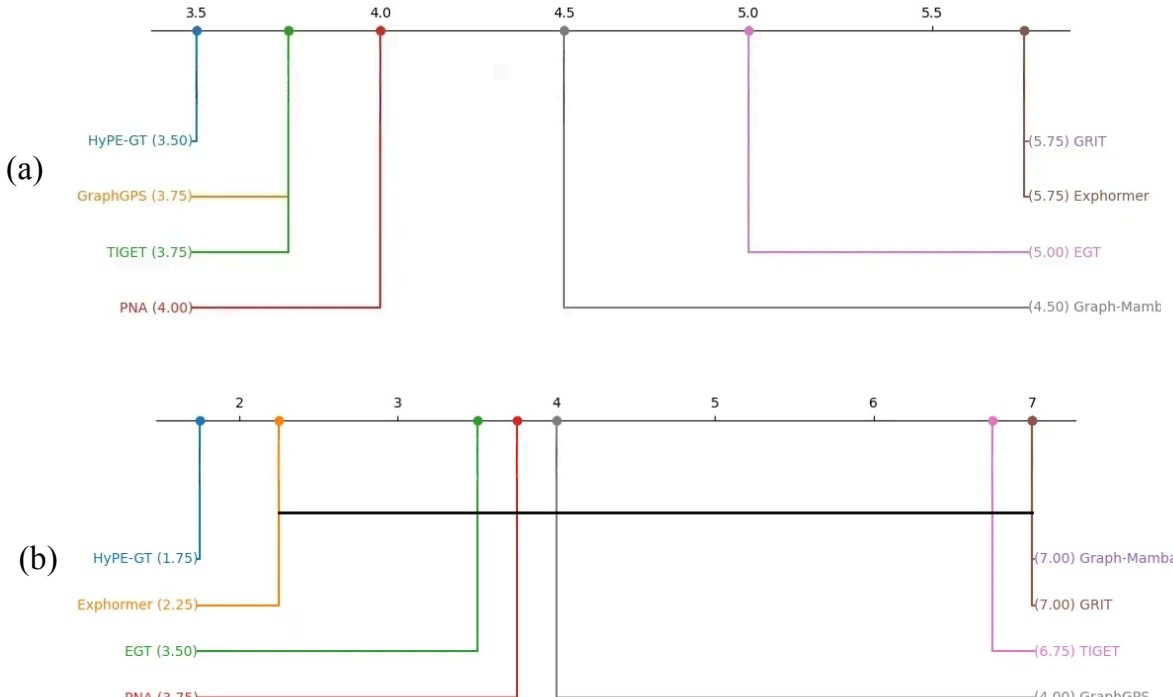

Figure 9: The critical difference diagrams are presented for Tables 1 and 2. The ranking of the top 8 contender methods is presented. The higher-ranked methods are placed from left to right. The ranks are mentioned on the horizontal line. HyPE-GT emerged as the sole winning method in both cases.

diagrams are a clear indicator of marking HyPE-GT as a winning method compared to other state-of-the-art methods.

## 11 Implementation Details and Numerical Stability

To ensure reproducibility and numerical stability of hyperbolic operations in HyPE-GT, we adopt the following implementation practices:

**Choice of Curvature.** HyPE-GT operates on the Poincaré ball and the Hyperboloid with constant negative curvature $c$. The curvature parameter $c$ is initialized to 1.0 and is optionally learnable during training. We apply gradient clipping to keep $c \in [1 \times 10^{-5}, 5.0]$ to prevent flat or contraction of curved spaces.

**Numerically Stable Riemannian Operations.** All hyperbolic operations (logarithmic map, exponential map, and Möbius addition) are implemented using numerically stable variants as follows:

$$\exp_0(x) = \tanh(\sqrt{c}\|x\|)\frac{x}{\sqrt{c}\|x\| + \varepsilon}, \quad \log_0(y) = \tanh^{-1}(\sqrt{c}\|y\|)\frac{y}{\sqrt{c}\|y\| + \varepsilon}$$

A small constant $\varepsilon = 1 \times 10^{-5}$ is added to avoid division by zero and to stabilize numerical computations. To prevent the numerical overflow, we employed the `torch.clamp(.)` function to limit the values within a fixed range.

**Parameter Initialization and Normalization.** The $L_2$-normalization is applied to both node and positional encodings. Linear transformations are initialized using `xavier_uniform` initialization to maintain balanced variance and stabilized training.

**Reproducibility.** We fix random seeds (`torch.manual_seed(42)`, `np.random.seed(42)`) and disable non-deterministic CuDNN operations (`torch.use_deterministic_algorithms(True)`) to ensure the reproducibility of the results. All experiments are conducted on identical hardware and with identical optimizer configurations across the runs. Furthermore, no `NaN` or `Inf` values were observed across training or inference runs. Gradient norms and curvature magnitudes remain stable throughout training (maximum relative change $< 0.02$).

Overall, these safeguards ensure that hyperbolic operations remain numerically stable and reproducible across all datasets. No instabilities were observed during training, confirming the robustness of the implementation.

## 12    Limitations and Applicability

HyPE-GT excels at capturing intricate hierarchical relationships within graphs by generating learnable positional encodings in hyperbolic space. This approach is particularly effective when dealing with input graphs that possess inherent hierarchical structures, as hyperbolic space provides superior encoding capabilities compared to Euclidean space due to its exponential growing nature. However, not all input graphs exhibit hierarchical characteristics. In such cases, positional encodings learned in Euclidean space may be more appropriate. For instance, the performance of HyPE-GT on PATTRN and CLUSTER datasets, which lack hierarchical components, shows slight degradation. Euclidean GT is preferably suitable for graphs with no exponentially growing components. Conversely, HyPE-GT performs commendably on the ogbg-molhiv and ogbg-molpcba datasets, which contain hierarchical structures. These results demonstrate that HyPE-GT is proficient in generating effective positional encodings that align well with hierarchical structures. Moreover, they underscore the framework's versatility and robustness in handling graphs with varying degrees of hierarchical complexity.

The performance of HyPE-GT can be further improved by tuning the space curvature. The optimal curvature value depends on the underlying datasets and chosen manifold type. In this work, a single value of curvature is considered to be shared across the nodes.

## 13    Ablation on Space Curvature

We conducted a detailed ablation study on space curvature for datasets such as PATTERN, MNIST, ogbg-molhiv, and ogbg-code2. The values of curvatures are chosen from the set $\{0.001, 0.01, 0.1, 1.0, 2.0, 5.0, 10.0\}$. The performance trends are illustrated in Figure 10. The performance trends indicate the best performance

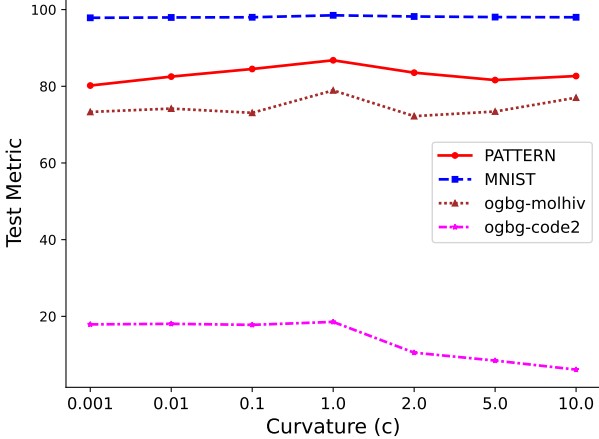

Figure 10: The performance metric is presented against a set of curvatures on four datasets. The optimal performance is mostly observed when $c = 1.0$.

Table 6: A comparative study of prevalent structural encoding strategies in Graph Transformers and their compatibility to the proposed hyperbolic positional encoding.

| Category | Representative Methods | Description | Compatible to HyPE-GT |
|---|---|---|---|
| Spectral Encodings | Laplacian eigenvectors (Dwivedi & Bresson, 2020) | Capture smooth global structure via eigen-decomposition of the graph Laplacian. | Yes. |
| Distance- or Shortest-path-based Encodings | Graphormer (Ying et al., 2021), SAN (Kreuzer et al., 2021) | Inject pairwise shortest-path distances or centrality measures as attention bias terms. | No |
| Random-walk-based Encodings | RWPE (Dwivedi et al., 2021), RRWP (Ma et al., 2023) | Capture local diffusion processes or transition probabilities derived from random walks. | Yes, but for RRWP, pairwise encodings are not compatible. |
| Structural Role or Sub-graph Encodings | SAT (Chen et al., 2022) | Encode structural roles, motifs, or subgraph identifiers to capture higher-order structural patterns. | Yes |

is achieved when $c = 1.0$. The performance gain is observed from a lower curvature (0.001) to a higher one. Notably, the performance of HYPE-GT degrades as the curvature value increases. The phenomena can be attributed to the space contraction due to high curvature. In our experiments, we pursue the strategy of learning curvatures across the hidden layers for the HyPE module (Chami et al., 2019). The initial curvature value is initialized to 1.0 for every dataset, and during training, the optimal space curvatures are learned. The benefits of learning curvatures alleviate the problem of setting curvature as a tunable hyperparameter.

## 14    HyPE-GT as a Framework

The initialization submodule of HyPE can serve as a plug-and-play solution for other absolute positional encodings, such as LapPE, RWPE, RRWP (Ma et al., 2023), and SAT (Chen et al., 2022). Thus, we conducted an experiment to validate the utility of HyPE as a flexible framework. For the existing positional encoding, we run the vanilla Graph Transformer on 8 datasets. Subsequently, we further run experiments by transforming the initial PEs through the HyPE module, and the corresponding results are provided in Table 7. Minor degradation of performance is noticeable in a few cases due to the non-hierarchical graphs, such as PATTERN and CLUSTER. On the other side, the performance gain of GT is impressive and underscores the role of HyPE as an agnostic solution for absolute positional encodings.

**Connection with Structural Encodings.** Structural encodings like node degrees, triangle or ring count (Bouritsas et al., 2022b), (Zhao et al., 2021) can be directly added or concatenated with the initial node features. In contrast, relative structural encodings like pair-wise shortest path distance are added with the self-attention matrix, unlike directly adding or concatenating with node features. Our framework HyPE, is predominantly compatible with the addition or concatenation of the structural encodings with the input node features. This is due to we transformed the PEs in hyperbolic space, reverting to Euclidean space, and subsequently, the learned PEs are fed to the Transformer block. Thus, incorporating relative structural encodings in the self-attention matrix will be theoretically incorrect and logically insignificant. Hence, HyPE is capable of accommodating absolute positional encodings. We further presented a compatibility study among the structural encodings with respect to our framework HyPE in Table 6.

## 15    Conclusion & Future Works

In this work, we proposed a novel framework called HyPE-GT to generate positional encodings in the hyperbolic space for Graph Transformers. Unlike the other existing methods, our framework can generate a set of positional encodings that offer diverse choices for solving downstream tasks. The generated PEs are learned either by HNN or HGCN-based architectures. The efficiency of HyPE-GT is also validated by performing several experiments on molecular graphs from benchmark datasets (Dwivedi et al., 2020) and OGB (Hu et al., 2020a) graph datasets and achieving impressive performances on the datasets. We provided the results of an exhaustive ablation study to substantiate the importance of each component of HyPE-GT. We also re-purpose the positional encodings to integrate with node features to boost the performance of deep graph neural networks applied on Co-author and Co-purchase datasets. Exploring hyperbolic spaces to learn positional encodings may be a potential avenue for future directions. Also, further investigation is required on the effectiveness of positional encodings in deeper GNNs to boost performance.

Table 7: The comparative performance of HyPE is presented with four types of positional encodings across 8 datasets. The best results are boldfaced. The performance of Graph Transformer improves when the positional encodings are transformed with HyPE, while in some cases, the performance marginally degrades. The '-' indicates the results are not available in the respective papers.

| PE | PATTERN | CLUSTER | MNIST | CIFAR10 |
|---|---|---|---|---|
| LapPE | $84.808 \pm 0.068$ | $73.169 \pm 0.622$ | $97.961 \pm 0.005$ | $68.419 \pm 0.013$ |
| LapPE + HyPE | $\mathbf{86.779 \pm 0.038}$ | $\mathbf{78.228 \pm 0.126}$ | $\mathbf{98.510 \pm 0.007}$ | $\mathbf{74.680 \pm 0.009}$ |
| RWPE | $85.148 \pm 0.051$ | $74.728 \pm 0.148$ | $\mathbf{97.810 \pm 0.015}$ | $69.104 \pm 0.006$ |
| RWPE + HyPE | $\mathbf{85.673 \pm 0.058}$ | $\mathbf{74.891 \pm 0.156}$ | $97.802 \pm 0.018$ | $\mathbf{69.547 \pm 0.009}$ |
| RRWP | $\mathbf{87.196 \pm 0.076}$ | $80.026 \pm 0.277$ | $\mathbf{98.108 \pm 0.111}$ | $76.468 \pm 0.881$ |
| RRWP + HyPE | $86.584 \pm 0.081$ | $\mathbf{80.213 \pm 0.295}$ | $97.995 \pm 0.124$ | $\mathbf{77.142 \pm 0.794}$ |
| SAT | $\mathbf{86.865 \pm 0.043}$ | $\mathbf{77.856 \pm 0.104}$ | – | – |
| SAT + HyPE | $86.792 \pm 0.051$ | $77.394 \pm 0.118$ | – | – |
| PE | ogbg-molhiv | ogbg-ppa | ogbg-molpcba | ogbg-code2 |
| LapPE | $0.7619 \pm 0.0141$ | $0.6864 \pm 0.0047$ | $0.1846 \pm 0.0158$ | $0.1738 \pm 0.0381$ |
| LapPE + HyPE | $\mathbf{0.7893 \pm 0.0005}$ | $\mathbf{0.7981 \pm 0.0043}$ | $\mathbf{0.2967 \pm 0.0079}$ | $\mathbf{74.680 \pm 0.009}$ |
| RWPE | $0.7689 \pm 0.0037$ | $0.6971 \pm 0.0032$ | $\mathbf{0.1879 \pm 0.0048}$ | $0.1721 \pm 0.0517$ |
| RWPE + HyPE | $\mathbf{0.7721 \pm 0.0041}$ | $\mathbf{0.6984 \pm 0.0038}$ | $0.1868 \pm 0.0053$ | $\mathbf{0.1804 \pm 0.0489}$ |
| RRWP | – | – | – | – |
| RRWP + HyPE | – | – | – | – |
| SAT | – | $0.7522 \pm 0.0056$ | – | $0.1937 \pm 0.0028$ |
| SAT + HyPE | – | $\mathbf{0.7681 \pm 0.0062}$ | – | $\mathbf{0.1942 \pm 0.0031}$ |

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

## A  Appendix

### A.1  Proofs

**Lemma 1** Consider an $n$-dimensional Poincarê Ball $\mathbb{B}^n$ of unit radius and unit curvature. Let us assume that two points $x, y \in \mathbb{B}^n$ and their distance by using Poincarê metric is $d_{\mathbb{B}}(x, y)$. If we apply the Euclidean metric to them, the distance will be $d_E(x, y)$. Then $\forall k \in (0, 1)$, we have $d_{\mathbb{B}}(x, y) \geq 2k d_E(x, y)$ if $d_E(x, y) \in [0, \frac{\sqrt{1-k^2}}{k}]$.

*Proof.* Assume Poincarê Ball $\mathbb{B}^n = \{x \in \mathbb{R}^n, ||x|| \leq 1\}$ is equipped with Poincarê metric and the distance between two points $x, y \in \mathbb{B}^n$ as following

$$d_{\mathbb{B}}(x, y) = \cosh^{-1}\left(1 + \frac{2||x - y||^2}{(1 - ||x||^2)(1 - ||y||^2)}\right) \tag{18}$$

The Euclidean norm between $x, y$ is

$$d_E(x, y) = ||x - y|| \tag{19}$$

The following can be written

$$\begin{aligned}
d_{\mathbb{B}}(x, y) &= \cosh^{-1}\left(1 + \frac{2||x - y||^2}{(1 - ||x||^2)(1 - ||y||^2)}\right) \\
&= 2\sinh^{-1}\left(\sqrt{\frac{\Delta(x, y)}{2}}\right)
\end{aligned} \tag{20}$$

where

$$\Delta(x, y) = \frac{2||x - y||^2}{(1 - ||x||^2)(1 - ||y||^2)}$$

We have considered the unit radius of the Poincarê Ball, $||x||, ||y|| \leq 1$. then $(1 - ||x||^2)(1 - ||y||^2) \leq 1$. Therefore, we have

$$\Delta(x, y) \geq 2||x - y||^2 \tag{21}$$

We have $\frac{d}{dx}(\sinh^{-1}(x)) = \frac{1}{\sqrt{1+x^2}} > 0$ which implies that $\sinh^{-1}(x)$ is an increasing function. Therefore, we have,

$$2\sinh^{-1}\left(\sqrt{\frac{\Delta(x,y)}{2}}\right) \geq 2\sinh^{-1}(||x-y||) \tag{22}$$
$$d_{\mathbb{B}}(x,y) \geq 2\sinh^{-1}(||x-y||)$$

We know that $\sinh^{-1}(z) = \ln(z + \sqrt{z^2 + 1})$. Let us consider the following function $\forall k \in \mathbb{R}$

$$f(z) = \ln(z + \sqrt{z^2 + 1}) - kz \tag{23}$$

Differentiating w.r.t. $z$, we get

$$f'(z) = \frac{1}{\sqrt{1+x^2}} - k \tag{24}$$

Also $f(0) = 0$ and the function is increasing when $f'(z) \geq 0$

$$f'(z) = \frac{1}{\sqrt{1+x^2}} - k \geq 0$$
$$\frac{1}{\sqrt{1+x^2}} \geq k \tag{25}$$
$$z \leq \frac{\sqrt{1-k^2}}{k}$$

If the above condition holds then $f(z)$ increases and non-negative $\forall z \in [0, \frac{\sqrt{1-k^2}}{k}]$. Then we have

$$\ln(z + \sqrt{z^2 + 1}) - kz \geq 0$$
$$\ln(z + \sqrt{z^2 + 1}) \geq kz \tag{26}$$
$$\sinh^{-1}(z) \geq kz \ \forall z \in [0, \frac{\sqrt{1-k^2}}{k}]$$

Applying the inequality to the distance between points $x, y$ and $\forall ||x-y|| \in [0, \frac{\sqrt{1-k^2}}{k}]$ we have,

$$\sinh^{-1}(||x-y||) \geq k||x-y||$$
$$2\sinh^{-1}(||x-y||) \geq 2k||x-y|| \tag{27}$$
$$d_{\mathbb{B}}(x,y) \geq 2k||x-y||$$

Therefore, the distance between any two points $x, y$ increases when the Poincarê metric is applied compared to the scaled Euclidean distance under certain conditions. $\qquad\square$

**Lemma 2** Consider an $n$-dimensional Hyperboloid space $\mathbb{H}^n$ of unit radius and unit curvature. Let us assume that two points $x, y \in \mathbb{H}^n$ and their distance by using Hyperboloid distance metric is $d_{\mathbb{H}}(x,y)$. If we apply the Euclidean metric to them, the distance will be $d_E(x,y)$. Then $\forall k \in [1, \infty)$, we have $d_{\mathbb{H}}(x,y) \geq \frac{k}{2}d_E(x,y) \ \forall d_E(x,y) \in [1, \sqrt[4]{\frac{1+k^2}{k^2}}]$.

*Proof.* Let us define the Hyperboloid space in the following way

$$\mathbb{H}_1^n = \{x \in \mathbb{R}^{n+1} : \langle x, x \rangle_{\mathcal{L}} = -1, x_0 > 0\}, \tag{28}$$

where $\langle x, y \rangle_{\mathcal{L}}$ denotes Minkowski inner product is defined as follows

$$\langle x, y \rangle_{\mathcal{L}} = -x_0 y_0 + x_1 y_1 + \cdots + x_n y_n$$

The distance between $x, y$ can be estimated as

$$
\begin{aligned}
d_{\mathbb{H}}(x, y) &= \cosh^{-1}(-\langle x, y \rangle_{\mathcal{L}}) \\
&= \cosh^{-1}(-(-x_0 y_0 + \sum_{i=1}^{n} x_i y_i)) \\
&= \cosh^{-1}(x_0 y_0 - \sum_{i=1}^{n} x_i y_i)
\end{aligned}
\tag{29}
$$

For two points $x, y$, following the condition of $\langle x, x \rangle_{\mathcal{L}} = 1$, we have,

$$
\begin{aligned}
\langle x, x \rangle_{\mathcal{L}} &= -x_0^2 + x_1^2 + \cdots x_n^2 = -1 \\
\langle y, y \rangle_{\mathcal{L}} &= -y_0^2 + y_1^2 + \cdots y_n^2 = -1
\end{aligned}
\tag{30}
$$

The Euclidean distance between $x, y$ can be expressed as

$$
\begin{aligned}
d_E(x, y) &= \sqrt{\sum_{i=0}^{n} (x_i - y_i)^2} \\
&= \sqrt{\sum_{i=0}^{n} x_i^2 + \sum_{i=0}^{n} y_i^2 - 2 \sum_{i=0}^{d} x_i y_i} \\
&= \sqrt{x_0^2 + \sum_{i=1}^{n} x_i^2 + y_0^2 + \sum_{i=1}^{n} x_i^2 - 2(x_0 y_0 + \sum_{i=1}^{n} x_i y_i)}
\end{aligned}
\tag{31}
$$

Using from Eq. 29 and Eq. 30, we have

$$
\begin{aligned}
d_E(x, y) &= \sqrt{2x_0^2 + 2y_0^2 - 2 - 2(2x_0 y_0 - \cosh(d_{\mathbb{H}}(x, y)))} \\
d_E^2(x, y) &= 2x_0^2 + 2y_0^2 - 2 - 4x_0 y_0 + 2\cosh(d_{\mathbb{H}}(x, y)) \\
d_E^2(x, y) &= 2((x_0 - y_0)^2 - 1) + 2\cosh(d_{\mathbb{H}}(x, y))
\end{aligned}
$$

Assuming that $(x_0 - y_0)^2 \leq 1$, then we can express

$$
\begin{aligned}
2\cosh(d_{\mathbb{H}}(x, y)) &\geq d_E^2(x, y) \\
d_{\mathbb{H}}(x, y) &\geq \cosh^{-1}\left(\frac{d_E^2(x, y)}{2}\right)
\end{aligned}
\tag{32}
$$

Consider the function $f(z) = \cosh^{-1}(z) - kz$ where $\cosh^{-1}(z) = \ln(z + \sqrt{z^2 - 1})$. Differentiating $f(z)$ w.r.t $z$, we get

$$
\begin{aligned}
f'(z) &= \frac{1}{\sqrt{z^2 - 1}} - k \geq 0 \\
\frac{1}{\sqrt{z^2 - 1}} &\geq k \\
z &\leq \frac{\sqrt{1 + k^2}}{k}
\end{aligned}
\tag{33}
$$

Also, $f(0) = 0$, and the function is increasing for the above condition. Therefore, we have

$$
\begin{aligned}
f(z) = \cosh^{-1}(z) - kz &\geq 0 \\
\cosh^{-1}(z) \geq kz \quad \forall z \in [1, \frac{\sqrt{1 + k^2}}{k}]
\end{aligned}
\tag{34}
$$

Applying the above inequality, finally, we have,

$$d_{\mathbb{H}}(x,y) \geq \cosh^{-1}\left(\frac{d_E^2(x,y)}{2}\right) \geq k\frac{d_E^2(x,y)}{2}$$

$$d_{\mathbb{H}}(x,y) \geq \frac{k}{2}d_E^2(x,y) \geq \frac{k}{2}d_E(x,y) \;\forall d_E^2(x,y) \in [1, \frac{\sqrt{1+k^2}}{k}] \tag{35}$$

$$d_{\mathbb{H}}(x,y) \geq \frac{k}{2}d_E(x,y) \;\forall\, d_E(x,y) \in [1, \sqrt[4]{\frac{1+k^2}{k^2}}]$$

Therefore, the distance between two points in the hyperboloid space will be greater than their scaled Euclidean distance under certain conditions. $\qquad\square$

**Theorem 1** Consider a pair of nodes 1 and 2 of a connected graph $\mathcal{G}$ whose degrees are $d_1$ and $d_2$ respectively. Their initialized positional encodings are $p_1, p_2 \in \mathbb{R}^d$. The Euclidean distance between them is estimated as $d_E(p_1, p_2)$. Suppose, $p_1, p_2$ are to be transformed by either HNN or HGCN with the underlying hyperbolic space as a $n$-dimensional Poincaré Ball $\mathbb{B}^n$ of unit radius and unit curvature, then we have the following:

1. **HNN** If the encodings are transformed by passing through an HNN of parameters $\Theta$. The transformed encodings are respectively $p_1^{\text{hyp}}$ and $p_2^{\text{hyp}}$ whose distance is $d_{\mathbb{B}}(p_1^{\text{hyp}}, p_2^{\text{hyp}})$, then $\exists\,\Theta'$ such that $d_{\mathbb{B}}(p_1^{\text{hyp}}, p_2^{\text{hyp}}) \geq k'||\Theta'||_F d_E(p_1, p_2)$ for some $k' \in (0,2)$ and $||\Theta'||_F \leq 1$.

2. **HGCN** If the encodings are transformed by passing through an HGCN of parameters $\Phi$. then $\exists\,\Phi'$ with $||\Phi'||_F \leq 1$ such that $d_{\mathbb{B}}(p_1^{\text{hyp}}, p_2^{\text{hyp}}) \geq \frac{k'}{d}||\Phi'||_F d_E(p_1, p_2)$ where $d = \max\{d_1, d_2\}$ for some $k' \in (0,2)$.

*Proof.* We will outline the complete proof for each of the parts.

**HNN** The Euclidean distance between two points $x, y$ is estimated as $d_E(x,y) = ||p_1 - p_2||$. We want to compute the distance between $p_1, p_2$ in the $\mathbb{B}_c^n$. Therefore, we will apply the exponential map to project the points in the manifold space. The exponential map for $\mathbb{B}_c^n$ at 0 is defined as following,

$$\exp_c^0(v) = \tanh(\sqrt{c}||v||)\frac{v}{c||v||}, \tag{36}$$

where $v$ is any point lying on the tangent space which resembles the locally linear space. As we considered the curvature to be 1, the exponential map can be reformulated as,

$$\exp_1^0(v) = \tanh(||v||)\frac{v}{||v||}, \tag{37}$$

From now on we will write as $\exp(v)$. The Möbius matrix-vector multiplication is defined as,

$$M^{\otimes_c}(x) = \frac{1}{\sqrt{c}}\tanh\left(\frac{||Mx||}{||x||}\tanh^{-1}(\sqrt{c}||x||)\right)\frac{Mx}{||Mx||}, \tag{38}$$

where $M \in \mathcal{M}_{m,n}(\mathbb{R})$ and $x \in \mathbb{B}_c^n$. Substituting $c = 1$, we get

$$M^{\otimes_1}(x) = \tanh\left(\frac{||Mx||}{||x||}\tanh^{-1}(||x||)\right)\frac{Mx}{||Mx||}, \tag{39}$$

If we pass the positional encodings through a hyperbolic neural network (HNN) with a trainable weight $\Theta \in \mathbb{R}^{d \times d'}$. Firstly, we mapped the encodings into the manifold space using the exponential map.

$$\hat{p}_1 = \exp(p_1) = \tanh(||p_1||)\frac{p_1}{||p_1||}$$

$$\hat{p}_2 = \exp(p_2) = \tanh(||p_2||)\frac{p_2}{||p_2||}$$

Replacing the scalar terms $\zeta_1 = \frac{\tanh(||p_1||)}{||p_1||}$ and $\zeta_2 = \frac{\tanh(||p_2||)}{||p_2||}$. Then, we have $\hat{p}_1 = \zeta_1 p_1$ and $\hat{p}_2 = \zeta_2 p_2$.

The encodings are now mapped on the manifold. The encodings are transformed by passing through HNN, then applying Möbius matrix-vector formula, we have

$$p_1^{\text{hyp}} = \Theta^{\otimes_1}(\hat{p}_1) = \tanh\left(\frac{||\hat{p}_1\Theta||}{||\hat{p}_1||}\tanh^{-1}(||\hat{p}_1||)\right)\frac{\hat{p}_1\Theta}{||\hat{p}_1\Theta||}$$

$$\text{Substituting } \hat{p}_1 = \zeta_1 p_1$$

$$= \tanh\left(\frac{||p_1\Theta||}{||p_1||}\tanh^{-1}(||\zeta_1 p_1||)\right)\frac{p_1\Theta}{||p_1\Theta||}$$

Similarly, we can write

$$p_2^{\text{hyp}} = \Theta^{\otimes_1}(\hat{p}_2) = \tanh\left(\frac{||p_2\Theta||}{||p_2||}\tanh^{-1}(||\zeta_2 p_2||)\right)\frac{p_2\Theta}{||p_2\Theta||}$$

Replacing the scalar terms as $\eta_1 = \frac{\tanh^{-1}(||\zeta_1 p_1||)}{||p_1||}$ and $\eta_2 = \frac{\tanh^{-1}(||\zeta_2 p_2||)}{||p_2||}$, we have $p_1^{\text{hyp}} = \tanh(\eta_1||p_1\Theta||)\frac{p_1\Theta}{||p_1\Theta||}$ and $p_2^{\text{hyp}} = \tanh(\eta_2||p_2\Theta||)\frac{p_2\Theta}{||p_2\Theta||}$. Further substituting the scalar terms as $\omega_1 = \frac{\tanh(\eta_1||p_1\Theta||)}{||p_1\Theta||}$ and $\omega_2 = \frac{\tanh(\eta_2||p_2\Theta||)}{||p_2\Theta||}$. Therefore, we can write as,

$$p_1^{\text{hyp}} = \omega_1 p_1 \Theta \qquad\qquad p_2^{\text{hyp}} = \omega_2 p_2 \Theta \tag{40}$$

Therefore, the length of the geodesic on the manifold will be,

$$d_{\mathbb{B}}(p_1^{\text{hyp}}, p_2^{\text{hyp}}) = \cosh^{-1}\left(1 + \frac{2||p_1^{\text{hyp}} - p_2^{\text{hyp}}||^2}{(1 - ||p_1^{\text{hyp}}||^2)(1 - ||p_2^{\text{hyp}}||^2)}\right)$$

$$= \cosh^{-1}\left(1 + \frac{2||\omega_1 p_1\Theta - \omega_2 p_2\Theta||^2}{(1 - ||\omega_1 p_1\Theta||^2)(1 - ||\omega_2 p_2\Theta||^2)}\right)$$

We know $\frac{\tanh(z)}{z} \leq 1 \forall z \in \mathbb{R}$. The $\frac{\tanh^{-1}(tz)}{z} \leq 1 \ \forall |t| \leq 1, |z| \leq 1$. Therefore, we can say that $\zeta_1, \zeta_2 \leq 1$. We use this notion to have $\eta_1, \eta_2 \leq 1$. Finally, $\omega_1, \omega_2 \leq 1 \ \forall z \in \mathbb{R}$.

Applying the inequalities, $||\omega_1 p_1 \Theta|| \leq \omega_1 ||p_1||||\Theta||_F$. As we considered unit radius $||p_1|| \leq 1$ and assumed a bounded Frobenius norm of the HNN parameters $||\Theta||_F \leq 1$. Thus we have $||\omega_1 p_1 \Theta|| \leq 1$ and in a similar way $||\omega_2 p_2 \Theta|| \leq 1$ which implies $(1 - ||\omega_1 p_1\Theta||^2)(1 - ||\omega_2 p_2\Theta||^2) \leq 1$.

$$d_{\mathbb{B}}(p_1^{\text{hyp}}, p_2^{\text{hyp}}) = 2\sinh^{-1}\left(\sqrt{\frac{\Delta(p_1^{\text{hyp}}, p_2^{\text{hyp}})}{2}}\right) \tag{41}$$

where

$$\Delta(p_1^{\text{hyp}}, p_2^{\text{hyp}}) = \frac{2||\omega_1 p_1\Theta - \omega_2 p_2\Theta||^2}{(1 - ||\omega_1 p_1\Theta||^2)(1 - ||\omega_2 p_2\Theta||^2)}$$

We have $\frac{d}{dx}(\sinh^{-1}(x)) = \frac{1}{\sqrt{1+x^2}} > 0$ which implies that $\sinh^{-1}(x)$ is an increasing function. Therefore, we have,

$$2\sinh^{-1}\left(\sqrt{\frac{\Delta(p_1^{\text{hyp}}, p_2^{\text{hyp}})}{2}}\right) \geq 2\sinh^{-1}(||p_1^{\text{hyp}} - p_2^{\text{hyp}}||)$$

$$d_{\mathbb{B}}(p_1^{\text{hyp}}, p_2^{\text{hyp}}) \geq 2\sinh^{-1}(||p_1^{\text{hyp}} - p_2^{\text{hyp}}||) \tag{42}$$

WLOG, we can assume $\omega_1 \geq \omega_2$ and apply properties of Euclidean norm. Then we have,

$$
\begin{aligned}
||p_1^{\text{hyp}} - p_2^{\text{hyp}}|| &= ||\omega_1 p_1 \Theta - \omega_2 p_2 \Theta|| \\
&\geq (||\omega_1 p_1 \Theta|| - ||\omega_2 p_2 \Theta||) \\
&\geq \omega_2 (||p_1 \Theta|| - ||p_2 \Theta||)
\end{aligned}
\tag{43}
$$

Again, we can write,

$$
\omega_2 (||p_1 \Theta|| - ||p_2 \Theta||) \leq \omega_2 (||p_1 \Theta - p_2 \Theta||)
\tag{44}
$$

Using the properties of the vector norms, we have,

$$
\omega_2 ||p_1 \Theta - p_2 \Theta|| \leq \omega_2 ||\Theta||_F ||p_1 - p_2||
\tag{45}
$$

Combining Eqn. 44 and 45, we have,

$$
\omega_2 ||\Theta||_F ||p_1 - p_2|| \geq \omega_2 (||p_1 \Theta|| - ||p_2 \Theta||)
\tag{46}
$$

Therefore, $\exists\, \Theta'$ with $||\Theta'||_F \leq 1$ such that,

$$
|p_1^{\text{hyp}} - p_2^{\text{hyp}}|| \geq \omega_2 ||\Theta'||_F ||p_1 - p_2|| \geq \omega_2 (||p_1 \Theta|| - ||p_2 \Theta||)
\tag{47}
$$

Now combining with the Eq. 42 and using the notion from Theorem 1, we have,

$$
\begin{aligned}
2 \sinh^{-1}(||p_1^{\text{hyp}} - p_2^{\text{hyp}}||) &\geq 2k ||p_1^{\text{hyp}} - p_2^{\text{hyp}}|| \\
\forall ||p_1^{\text{hyp}} - p_2^{\text{hyp}}|| &\in [0, \frac{\sqrt{1-k^2}}{k}] \\
d_{\mathbb{B}}(p_1^{\text{hyp}}, p_2^{\text{hyp}}) &\geq 2k\omega_2 ||\Theta'||_F ||p_1 - p_2|| \\
&= k' ||\Theta'||_F ||p_1 - p_2|| \\
&\text{where } k' = 2k\omega_2 \text{ with } k' \in (0, 2)
\end{aligned}
\tag{48}
$$

**HGCN** If we employ an HGCN architecture with trainable parameters $\Phi$, we need to incorporate normalized adjacency matrix $\tilde{A}$ into the positional encodings. The transformed encoding is mapped into the manifold's tangent space using the logarithmic map to enable multiplication with $\tilde{A}$. The logarithmic or simply log map at 0 position is defined as

$$
\log_0^c(v) = \tanh^{-1}(\sqrt{c}||v||) \frac{v}{\sqrt{c}||v||}
\tag{49}
$$

Applying the log map with $c = 1$, the encodings are transformed into the tangent space as following

$$
\begin{aligned}
p_1^T &= \tanh^{-1}(||p_1^{\text{hyp}}||) \frac{p_1^{\text{hyp}}}{||p_1^{\text{hyp}}||} \\
p_2^T &= \tanh^{-1}(||p_2^{\text{hyp}}||) \frac{p_2^{\text{hyp}}}{||p_2^{\text{hyp}}||}
\end{aligned}
\tag{50}
$$

Consider $\eta_1' = \frac{\tanh^{-1}(||p_1^{\text{hyp}}||)}{||p_1^{\text{hyp}}||}$ and $\eta_2' = \frac{\tanh^{-1}(||p_2^{\text{hyp}}||)}{||p_2^{\text{hyp}}||}$ with $\eta_1', \eta_2' \geq 1$. Therefore, we can express $p_1^T = \eta_1' p_1^{\text{hyp}}$ and $p_2^T = \eta_2' p_2^{\text{hyp}}$. The adjacency matrix is applied to the transformed encodings which result in $p_1^A = \frac{1}{d_1} \sum_{j \in N(1) \cup p_1^T} p_j^T$ and $p_2^A = \frac{1}{d_2} \sum_{j \in N(2) \cup p_2^T} p_j^T$ where $d_1, d_2$ are the degrees for node 1 and 2 respectively with $p_j^T$ are the corresponding neighbors' features. Suppose, $\exists\, C_1, C_2 \in \mathbb{R}$ such that,

$$
||p_1^A|| \geq \frac{1}{d_1} C_1 ||p_1^T|| \qquad\qquad ||p_2^A|| \geq \frac{1}{d_2} C_2 ||p_2^T||
\tag{51}
$$

From Eq 40, we have $||p_1^A|| \geq \frac{1}{d_1} C_1 \eta_1' ||p_1^{\text{hyp}}|| = \frac{1}{d_1} C_1 \eta_1' \omega_1 ||p_1 \Phi|| = \frac{1}{d_1} \omega_1' ||p_1 \Phi||$ where $\omega_1' = C_1 \eta_1' \omega_1$. Similarly, we have $||p_2^A|| \geq \frac{1}{d_2} \omega_2' ||p_2 \Phi||$ where $\omega_2' = C_2 \eta_2' \omega_2$. As $\omega_1, \omega_2 \leq 1$ with the assumption of

Again, we must apply the exponential map to revert the embeddings to the Poincarê Ball.

$$p_1^{\text{Ah}} = \tanh(||p_1^A||)\frac{p_1^A}{||p_1^A||}$$
$$p_2^{\text{Ah}} = \tanh(||p_2^A||)\frac{p_2^A}{||p_2^A||} \tag{52}$$

Let us assume that $\tau_1 = \frac{\tanh(||p_1^A||)}{||p_1^A||}$ and $\tau_2 = \frac{\tanh(||p_2^A||)}{||p_2^A||}$ with $\tau_1, \tau_2 \le 1$. Therefore, $p_1^{Ah} = \tau_1 p_1^A$ and $p_2^{Ah} = \tau_1 p_2^A$.

Similarly, we have $p_2^{Ah} = \frac{1}{d_2}\tau_2 p_2^A \ge \frac{1}{d_2}\tau_2 \omega_2' p_2 \Phi = \frac{1}{d_2}\tau_2' p_2 \Phi$ where $\tau_2' = \tau_2 \omega_2'$.

The distance between $p_1^{Ah}$ and $p_2^{Ah}$ is

$$d_{\mathbb{B}}(p_1^{Ah}, p_2^{Ah}) = \cosh^{-1}\left(1 + \frac{2||p_1^{Ah} - p_2^{Ah}||^2}{(1 - ||p_1^{Ah}||^2)(1 - ||p_2^{Ah}||^2)}\right)$$
$$= \cosh^{-1}\left(1 + \frac{2||\frac{1}{d_1}\tau_1' p_1 \Phi - \frac{1}{d_2}\tau_2' p_2 \Phi||^2}{(1 - ||\frac{1}{d_1}\tau_1' p_1 \Phi||^2)(1 - ||\frac{1}{d_2}\tau_2' p_2 \Phi||^2)}\right) \tag{53}$$

We have,

$$||p_1^{Ah}|| = \tau_1||p_1^A|| = \frac{\tau_1}{d_1}||\sum_{j \in N(1) \cup p_1^T} p_j^T|| \le \frac{\tau_1}{d_1}\sum_{j \in N(1) \cup p_1^T}||p_j^T|| \le \frac{\tau_1}{d_1}\sum_{j \in N(1) \cup p_1^T}||p_j^T||$$
$$\text{where } \eta_j' = \frac{\tanh^{-1}(||p_j^{\text{hyp}}||)}{||p_j^{\text{hyp}}||} = \frac{\tau_1}{d_1}\sum_{j \in N(1) \cup p_1^T}\eta_j'||p_j^{\text{hyp}}|| \le \frac{\tau_1}{d_1}\sum_{j \in N(1) \cup p_1^T}\eta_{\max}' = \tau_1 \eta_{\max}' \tag{54}$$

Assuming $\tau_1 \eta_{\max}' \le 1$, then we have $||p_1^{Ah}|| \le 1$. Similarly, $||p_2^{Ah}|| \le 1$. Using these inequalities, we have $(1 - ||p_1^{Ah}||^2)(1 - ||p_2^{Ah}||^2) \le 1$. We can express the following

$$d_{\mathbb{B}}(p_1^{Ah}, p_2^{Ah}) \ge 2\sinh^{-1}(||p_1^{Ah} - p_2^{Ah}||) \tag{55}$$

Now we can write the following,

$$||p_1^{Ah} - p_2^{Ah}|| = ||\tau_1 p_1^A - \tau_2 p_2^A|| \ge \tau_1||p_1^A|| - \tau_2||p_2^A|| \ge \frac{1}{d_1}\tau_1 \omega_1'||p_1 \Phi|| - \frac{1}{d_2}\tau_2 \omega_2'||p_2 \Phi|| \ge ||\frac{\tau_1'}{d_1}p_1 \Phi|| - ||\frac{\tau_2'}{d_2}p_2 \Phi||, \tag{56}$$

where $\tau_1' = \tau_1 \omega_1', \tau_2' = \tau_2 \omega_2'$. Using the properties of Euclidean vector norms, we have,

$$||\frac{\tau_1'}{d_1}p_1 \Phi - \frac{\tau_2'}{d_2}p_2 \Phi|| \ge ||\frac{\tau_1'}{d_1}p_1 \Phi|| - ||\frac{\tau_2'}{d_2}p_2 \Phi||$$
$$||\frac{\tau_1'}{d_1}p_1 \Phi - \frac{\tau_2'}{d_2}p_2 \Phi|| \le ||\Phi||_F||\frac{\tau_1'}{d_1}p_1 - \frac{\tau_2'}{d_2}p_2|| \tag{57}$$

Combining Eqn. 56 and 57, we have,

$$||\Phi||_F||\frac{\tau_1'}{d_1}p_1 - \frac{\tau_2'}{d_2}p_2|| \ge ||\frac{\tau_1'}{d_1}p_1 \Phi|| - ||\frac{\tau_2'}{d_2}p_2 \Phi|| \tag{58}$$

Therefore, $\exists \Phi'$ with $||\Phi'|| \le 1$, such that,

$$||p_1^{Ah} - p_2^{Ah}|| \ge ||\Phi||_F||\frac{\tau_1'}{d_1}p_1 - \frac{\tau_2'}{d_2}p_2|| \ge ||\frac{\tau_1'}{d_1}p_1 \Phi|| - ||\frac{\tau_2'}{d_2}p_2 \Phi|| \tag{59}$$

WLOG, we can assume $\tau_1' \ge \tau_2'$, then we have,

$$||p_1^{Ah} - p_2^{Ah}|| \ge \tau_2'||\Phi||_F||\frac{p_1}{d_1} - \frac{p_2}{d_2}|| \tag{60}$$

If $d = \max\{d_1, d_2\}$, then we have,

$$
\begin{aligned}
||p_1^{Ah} - p_2^{Ah}|| &\geq \tau_2' ||\Phi|| ||\frac{p_1}{d_1} - \frac{p_2}{d_2}|| \\
&\geq \frac{\tau_2' ||\Phi||_F}{d} ||p_1 - p_2||
\end{aligned}
\tag{61}
$$

Therefore, from Lemma 1, we can express

$$
\begin{aligned}
d_{\mathbb{B}}(p_1^{Ah}, p_2^{Ah}) &\geq 2k ||p_1^{Ah} - p_2^{Ah}|| \; \forall k \in [0, \frac{\sqrt{1-k^2}}{k}] \\
&\geq 2k \frac{\tau_2' ||\Phi||_F}{d} ||p_1 - p_2|| \\
&= \frac{k'}{d} ||\Phi||_F ||p_1 - p_2||,
\end{aligned}
\tag{62}
$$

where $k' = 2k\tau_2'$ with $k' \in (0, 2)$. The above inequality illustrates that the distance of the two points lying on the Poincarê Ball is greater than the distance of the same when lying on the Euclidean space under certain conditions. Furthermore, the inequality depends on the maximum degree of the pair of nodes in the graph.

If $d$ increases, the distance on the Poincarê Ball also increases, which underscores the more distinctive positional encodings for the nodes having higher importance in the graph. □

**Theorem 2** Consider a pair of nodes 1 and 2 of a connected graph $\mathcal{G}$ whose degrees are $d_1$ and $d_2$ respectively. Their initialized positional encodings are $p_1, p_2 \in \mathbb{R}^d$. The Euclidean distance between them is estimated as $d_E(p_1, p_2)$. Suppose, $p_1, p_2$ are to be transformed y either HNN or HGCN with the underlying hyperbolic space as a $n$-dimensional Hyperboloid model $\mathbb{H}^n$ of unit radius and unit curvature, then we have the following:

1. **HNN** If the encodings are transformed by passing through an HNN of parameters $\Theta$. The transformed encodings are respectively $p_1^{\text{hyp}}$ and $p_2^{\text{hyp}}$ whose distance is $d_{\mathbb{H}}(p_1^{\text{hyp}}, p_2^{\text{hyp}})$, then $\exists \Theta'$ such that $d_{\mathbb{H}}(p_1^{\text{hyp}}, p_2^{\text{hyp}}) \geq \frac{k'}{2} ||\Theta'||_F d_E(p_1, p_2)$ for some $k' \in [1, \infty)$ and $||\Theta'||_F \leq 1$.

2. **HGCN** If the encodings are transformed by passing through an HGCN of parameters $\Phi$. then there exists a $\Phi'$ with $||\Phi'||_F \leq 1$ such that $d_{\mathbb{B}}(p_1^{\text{hyp}}, p_2^{\text{hyp}}) \geq \frac{k''}{2d} ||\Phi'||_F d_E(p_1, p_2)$ where $d = \max\{d_1, d_2\}$ for some $k' \in [1, \infty)$.

*Proof.* We will provide proof for each of the two parts.

**HNN** For any $v \in \mathcal{T}_0 \mathbb{H}_1^n$ and $y \in \mathbb{H}_1^n$, then the exponential and logarithmic map of $\mathbb{H}_1^n$ at $x = 0$ with curvature $c = 1$ can be expressed as follows,

$$
\begin{aligned}
\exp_0^1(v) &= \sinh(||v||_{\mathcal{L}}) \frac{v}{||v||_{\mathcal{L}}} \\
\log_0^1(y) &= \cosh^{-1}(1 + \epsilon) \frac{y}{||y||_{\mathcal{L}}},
\end{aligned}
\tag{63}
$$

where $\epsilon$ is a significantly non-negative real quantity. If $y_0 = 0$ then $||y||_{\mathcal{L}}$ is equivalent to $||y||_2$. Consider two initialized positional encodings $p_1, p_2 \in \mathcal{T}_o \mathbb{H}_1^n$ lying in the tangent space of $x = 0$ which resembles locally to the Euclidean space. If an HNN transforms the encodings with trainable parameters $\Theta$, then the transformed encodings can be represented as,

$$
\hat{p}_1 = p_1 \Theta \qquad\qquad \hat{p}_2 = p_2 \Theta
\tag{64}
$$

After the transformation, the encodings are mapped to the Hyperboloid space by applying the exponential map.

$$
\begin{aligned}
p_1^{\text{hyp}} &= \exp(\hat{p}_1) = \sinh(||\hat{p}_1||) \frac{\hat{p}_1}{||\hat{p}_1||} \\
p_2^{\text{hyp}} &= \exp(\hat{p}_2) = \sinh(||\hat{p}_2||) \frac{\hat{p}_2}{||\hat{p}_2||}
\end{aligned}
\tag{65}
$$

Assuming the scalar terms $\gamma_1 = \frac{\sinh(||\hat{p}_1||)}{||\hat{p}_1||}$ and $\gamma_2 = \frac{\sinh(||\hat{p}_2||)}{||\hat{p}_2||}$, we have $p_1^{\text{hyp}} = \gamma_1 p_1 \Theta$ and $p_2^{\text{hyp}} = \gamma_2 p_2 \Theta$. The distance between $p_1^{\text{hyp}}$ and $p_2^{\text{hyp}}$ is

$$d_{\mathbb{H}}(p_1^{\text{hyp}}, p_2^{\text{hyp}}) = \cosh^{-1}(-\langle p_1^{\text{hyp}}, p_2^{\text{hyp}} \rangle_{\mathcal{L}}) \tag{66}$$

From Lemma 2, we have the following inequality

$$\begin{aligned}
d_{\mathbb{H}}(p_1^{\text{hyp}}, p_2^{\text{hyp}}) &\geq \frac{k}{2} d_E(p_1^{\text{hyp}}, p_2^{\text{hyp}}) \\
&\qquad \forall\, d_E(p_1^{\text{hyp}}, p_2^{\text{hyp}}) \in [1, \sqrt[4]{\frac{1+k^2}{k^2}}] \\
&= \frac{k}{2} d_E(\gamma_1 \hat{p}_1, \gamma_2 \hat{p}_2) \\
&= \frac{k}{2} d_E(\gamma_1 p_1 \Theta, \gamma_2 p_2 \Theta) \\
&= \frac{k}{2} ||\gamma_1 p_1 \Theta - \gamma_2 p_2 \Theta||
\end{aligned} \tag{67}$$

Using the properties of Euclidean norm, we have,

$$\begin{aligned}
\frac{k}{2} ||\gamma_1 p_1 \Theta - \gamma_2 p_2 \Theta|| &\leq \frac{k}{2} ||\Theta||_F ||\gamma_1 p_1 - \gamma_2 p_2|| \\
\frac{k}{2} ||\gamma_1 p_1 \Theta - \gamma_2 p_2 \Theta|| &\geq \frac{k}{2} (||\gamma_1 p_1 \Theta|| - ||\gamma_2 p_2 \Theta||)
\end{aligned} \tag{68}$$

Therefore, $\exists\, \Theta'$ such that $||\Theta'||_F \leq 1$ with satisfies the following

$$d_{\mathbb{H}}(p_1^{\text{hyp}}, p_2^{\text{hyp}}) \geq \frac{k}{2} ||\Theta'||_F ||\gamma_1 p_1 - \gamma_2 p_2|| \geq \frac{k}{2} ||\gamma_1 p_1 \Theta - \gamma_2 p_2 \Theta|| \tag{69}$$

WLOG, we can assume $\gamma_1 \geq \gamma_2$, we have,

$$\begin{aligned}
d_{\mathbb{H}}(p_1^{\text{hyp}}, p_2^{\text{hyp}}) &\geq \frac{k\gamma_2}{2} ||\Theta'||_F ||p_1 - p_2|| \\
&= \frac{k'}{2} ||p_1 - p_2||,
\end{aligned} \tag{70}$$

where $k' = k\gamma_2$

**HGCN** If we want to transform the positional encodings with HGCN with trainable parameters $\Phi$, then we need to first feed the encodings to the dense layer. Now applying $\Phi$, the transformed encodings can be expressed as,

$$\hat{p}_1 = p_1 \Phi \qquad\qquad \hat{p}_2 = p_2 \Phi \tag{71}$$

The adjacency matrix is applied to the transformed encodings which result in $p_1^A = \frac{1}{d_1} \sum\limits_{j \in N(1) \cup \hat{p}_1} \hat{p}_j$ and $p_2^A = \frac{1}{d_2} \sum\limits_{j \in N(2) \cup \hat{p}_2} \hat{p}_j$ where $d_1, d_2$ are the degrees for node 1 and 2 respectively with $\hat{p}_j$ are the corresponding neighbors' features. Now $\exists\, C_1, C_2$ such that,

$$||p_1^A|| \geq \frac{1}{d_1} C_1 ||\hat{p}_1|| \qquad\qquad ||p_2^A|| \geq \frac{1}{d_2} C_2 ||\hat{p}_2|| \tag{72}$$

Now the updated positional encodings are reverted to the Hyperboloid space by applying the following exponential map

$$\begin{aligned}
p_1^{\text{hyp}} &= \exp(p_1^A) = \sinh(||p_1^A||) \frac{p_1^A}{||p_1^A||} \\
p_2^{\text{hyp}} &= \exp(p_2^A) = \sinh(||p_2^A||) \frac{p_2^A}{||p_2^A||}
\end{aligned} \tag{73}$$

Replacing the scalar terms as $\alpha_1 = \frac{\sinh(\|p_1^A\|)}{\|p_1^A\|}$ and $\alpha_2 = \frac{\sinh(\|p_2^A\|)}{\|p_2^A\|}$, we have $p_1^{\mathrm{hyp}} = \alpha_1 p_1^A$ and $p_2^{\mathrm{hyp}} = \alpha_2 p_2^A$. Therefore, the distance between $p_1^{\mathrm{hyp}}$ and $p_2^{\mathrm{hyp}}$ is

$$d_{\mathbb{B}}(p_1^{\mathrm{hyp}}, p_2^{\mathrm{hyp}}) = \cosh^{-1}(-\langle p_1^{\mathrm{hyp}}, p_2^{\mathrm{hyp}}\rangle_{\mathcal{L}}). \tag{74}$$

From Theorem 2, we have the following inequality

$$
\begin{aligned}
d_{\mathbb{H}}(p_1^{\mathrm{hyp}}, p_2^{\mathrm{hyp}}) &\geq \frac{k}{2} d_E(p_1^{\mathrm{hyp}}, p_2^{\mathrm{hyp}}) \\
&\qquad \forall\ d_E(p_1^{\mathrm{hyp}}, p_2^{\mathrm{hyp}}) \in [1, \sqrt[4]{\frac{1+k^2}{k^2}}] \\
&= \frac{k}{2} d_E(\alpha_1 p_1^A, \alpha_2 p_2^A) \\
&= \frac{k}{2} \|\alpha_1 p_1^A - \alpha_2 p_2^A\| \\
&\geq \frac{k}{2} (\alpha_1 \|p_1^A\| - \alpha_2 \|p_2^A\|) \\
&\geq \frac{k}{2} \left(\frac{\alpha_1 C_1}{d_1} \|\hat{p}_1\| - \frac{\alpha_2 C_2}{d_2} \|\hat{p}_2\|\right) \\
&= \frac{k}{2} \left(\frac{\alpha_1 C_1}{d_1} \|p_1 \Phi\| - \frac{\alpha_2 C_2}{d_2} \|p_2 \Phi\|\right)
\end{aligned}
\tag{75}
$$

WLOG, we can assume $\alpha_1 C_1 \geq \alpha_2 C_2$, then

$$d_{\mathbb{H}}(p_1^{\mathrm{hyp}}, p_2^{\mathrm{hyp}}) \geq \frac{k\alpha_2 C_2}{2} \left(\|\frac{p_1 \Phi}{d_1}\| - \|\frac{p_2 \Phi}{d_2}\|\right) \tag{76}$$

Using the properties of the Euclidean vector norm we have,

$$
\begin{aligned}
\|\frac{p_1 \Phi}{d_1} - \frac{p_2 \Phi}{d_2}\| &\leq \|\Phi\|_F \|\frac{p_1}{d_1} - \frac{p_2}{d_2}\| \\
\|\frac{p_1 \Phi}{d_1} - \frac{p_2 \Phi}{d_2}\| &\geq \|\frac{p_1 \Phi}{d_1}\| - \|\frac{p_2 \Phi}{d_2}\|
\end{aligned}
\tag{77}
$$

Therefore, $\exists\ \Phi'$ with $\|\Phi'\|_F \leq 1$ such that

$$
\begin{aligned}
d_{\mathbb{H}}(p_1^{\mathrm{hyp}}, p_2^{\mathrm{hyp}}) &\geq \frac{k\alpha_2 C_2}{2} \|\Phi'\|_F \|\frac{p_1}{d_1} - \frac{p_2}{d_2}\| \\
&\geq \frac{k\alpha_2 C_2}{2d} \|\Phi'\|_F \|p_1 - p_2\| \\
&\qquad \text{where } d = \max\{d_1, d_2\} \\
&= \frac{k'}{2d} \|\Phi'\|_F \|p_1 - p_2\|
\end{aligned}
\tag{78}
$$

Therefore, we have shown that an HGCN architecture exists such that the distance between two positional encodings increases compared to the same in Euclidean space. □

**Lemma 3** (Local Isometry of the Logarithmic Map). *Let $(\mathbb{B}_c^n, g^H)$ denote the n-dimensional Poincaré ball model of curvature $-c < 0$, and let $\log_0 : \mathbb{B}_c^n \to T_0 \mathbb{B}_c^n \cong \mathbb{R}^n$ denote the logarithmic map at the origin. For any two points $y_i, y_j \in \mathbb{B}_c^n$ satisfying $\|y_i\|, \|y_j\| \ll 1/\sqrt{c}$, the hyperbolic distance between them can be approximated by the Euclidean distance between their logarithmic images as follows,*

$$d_{\mathbb{H}}(y_i, y_j) = \|\log_0(y_i) - \log_0(y_j)\|_2 + \mathcal{O}(c\|y_i\|^3 + c\|y_j\|^3).$$

*Proof.* Let $y_i, y_j \in \mathbb{B}_c^n$ be two points on the Poincaré ball model of curvature $-c < 0$. The distance between them on the manifold is given by:

$$d_{\mathbb{H}}(y_i, y_j) = \frac{2}{\sqrt{c}} \tanh^{-1}\left(\sqrt{c}\,\|-y_i \oplus y_j\|\right),$$

where $\oplus$ denotes Möbius addition defined as:

$$x \oplus y = \frac{(1 + 2c\langle x, y\rangle + c\|y\|^2)x + (1 - c\|x\|^2)y}{1 + 2c\langle x, y\rangle + c^2\|x\|^2\|y\|^2}.$$

Assume both norms $\|y_i\|$ and $\|y_j\|$ are bounded relative to the curvature radius, i.e., $\|y_i\|, \|y_j\| \ll \frac{1}{\sqrt{c}}$.

Under this assumption, the denominator in the Möbius addition formula can be approximated by $1 + \mathcal{O}(c\|y_i\|^2)$, and the numerator can be expanded to first order in $c$:

$$(-y_i) \oplus y_j = (-y_i + y_j) + \mathcal{O}(c\|y_i\|^3 + c\|y_j\|^3).$$

Thus,

$$\|(-y_i) \oplus y_j\| = \|y_i - y_j\| + \mathcal{O}(c\|y_i\|^3 + c\|y_j\|^3).$$

Substituting the newer formula of the distance yields:

$$d_{\mathbb{H}}(y_i, y_j) = \frac{2}{\sqrt{c}} \tanh^{-1}\left(\sqrt{c}\|y_i - y_j\|\right) + \mathcal{O}(c\|y_i\|^3 + c\|y_j\|^3).$$

Using the Taylor series expansion of $\tanh^{-1}(z) = z + \frac{z^3}{3} + \mathcal{O}(z^5)$, we obtain:

$$d_{\mathbb{H}}(y_i, y_j) = 2\|y_i - y_j\| + \frac{2c\|y_i - y_j\|^3}{3} + \mathcal{O}(c^2\|y_i - y_j\|^5). \tag{79}$$

The logarithmic map at the origin is given by:

$$\log_0(y) = \frac{1}{\sqrt{c}} \tanh^{-1}(\sqrt{c}\|y\|)\frac{y}{\|y\|}.$$

For small $\|y\|$, we have $\tanh^{-1}(\sqrt{c}\|y\|) = \sqrt{c}\|y\| + \frac{c^{3/2}\|y\|^3}{3} + \mathcal{O}(c^{5/2}\|y\|^5)$. Substituting this gives:

$$\log_0(y) = y + \frac{c\|y\|^2}{3}y + \mathcal{O}(c^2\|y\|^4).$$

Let $\Delta = y_i - y_j$. Then, we have,

$$\begin{aligned}
\|\log_0(y_i) - \log_0(y_j)\| &= \|y_i - y_j + \tfrac{c}{3}(\|y_i\|^2 y_i - \|y_j\|^2 y_j)\| + \mathcal{O}(c^2\|y\|^4) \\
&= \|y_i - y_j\| + \mathcal{O}(c\|y_i\|^3 + c\|y_j\|^3).
\end{aligned} \tag{80}$$

From Eq. 79 and Eq. 80, we have the following expression.

$$d_{\mathbb{H}}(y_i, y_j) = \|\log_0(y_i) - \log_0(y_j)\|_2 + \mathcal{O}(c\|y_i\|^3 + c\|y_j\|^3). \tag{81}$$

Thus, the pairwise distance in the Euclidean tangent space differs from the hyperbolic distance only by a small curvature-dependent term. A similar argument can be established for the Hyperboloid model due their isometric correspondence with the Poincaré ball as per *Killing-Hopf Theorem* (Lang, 1995).

$\square$

## B  Details of the Datasets

The details of the datasets involved in the experiments are provided as follows,

Table 8: Details of the datasets from (Dwivedi et al., 2020) and (Hu et al., 2020a)

| Name | #Graphs | Avg # Nodes | Avg # Edges | Task | Directed | Metric |
|------|---------|-------------|-------------|------|----------|--------|
| PATTERN | 14000 | 118.9 | $3,039.3$ | binary classif. | No | Accuracy |
| CLUSTER | 12000 | 117.2 | $2,150.9$ | 6-class classif. | No | Accuracy |
| MNIST | $70,000$ | 70.6 | 564.5 | 10-class classification | Yes | Accuracy |
| CIFAR10 | $60,000$ | 117.6 | 941.1 | 10-class classification | Yes | Accuracy |
| ogbg-molhiv | 41127 | 25.5 | 27.5 | binary classif. | No | AUROC |
| ogbg-ppa | $158,100$ | 243.4 | $2,266.1$ | 37-task classic. | No | Accuracy |
| ogbg-molpcba | $437,929$ | 26.0 | 28.1 | 128-task classif. | No | Avg. Precision |
| ogbg-code2 | $452,741$ | 125.2 | 124.2 | 5 token sequence | No | F1 score |

Table 9: Details of the Co-author and Co-purchase datasets

| Dataset | Nodes | Edges | Features | Classes |
|---------|-------|-------|----------|---------|
| Amazon photo | 13752 | 491722 | 10 | 767 |
| Amazon Computers | 7650 | 238162 | 8 | 745 |
| Coauthor CS | 18333 | 81894 | 15 | 6805 |
| Coauthor Physics | 34493 | 495924 | 5 | 8415 |

**PATTERN and CLUSTER** are molecular datasets generated from Stochastic Block Model (Abbe, 2018). The prediction task here is an inductive node-level classification. In PATTERN the task is to identify which nodes in a graph belong to one of 100 different sub-graph patterns which were randomly generated with different SBM parameters. In CLUSTER, every graph is composed of 6 SBM-generated clusters, each drawn from the same distribution, with only a single node per cluster containing a unique cluster ID. Our target is predict the cluster ID of the nodes.

**MNIST and CIFAR10** are generated from image classification datasets of similar names. Superpixel datasets are constructed by an 8 nearest-neighbor graph of SLIC superpixels for each image. The 10-class classification tasks and standard dataset splits follow the original image classification datasets, i.e., for MNIST 55K/5K/10K and CIFAR10 45K/5K/10K train/validation/test graphs.

**ogbg-molhiv and ogbg-molpcba** are molecular property prediction datasets designed by OGB from MoleculeNet. The molecules are represented by the nodes (atoms) and edges (bonds). The node and edge features are generated from a similar source, which represents chemo-physical properties. The prediction task of ogbg-molhiv is the binary classification of the molecule's suitability for combating the replication of HIV. On the other hand, ogbg-molpcba, derived from PubChem BioAssay, is tasked to predict the results of 128 bioassays in multi-task binary classification.
setting.

**ogbg-ppa** (CC-0 license) consists of protein-protein association (PPA) networks derived from 1581 species categorized into 37 taxonomic groups. Nodes represent the proteins and edges are poised to encode the normalized level of 7 different associations between that pair of proteins. The target task is to classify to one of the 37 groups of the PPA network.

**ogbg-code2** (MIT License) is comprised of abstract syntax trees (ASTs) constructed from the source code of functions written in Python. The task is to predict the first 5 subtokens of the original function's name. A small number of these ASTs are much larger than the average size in the dataset. Therefore we truncated ASTs with over 1000 nodes and kept the first 1000 nodes according to their depth in the AST. The processing only impacted 2521 (0.5%) graphs in the entire dataset.

**Co-authorship datasets** Coauthor CS and Coauthor Physics are two co-authorship networks (Shchur et al., 2018). Nodes represent authors and edges exist between them if they co-authored a paper. The features of the nodes represent the keywords related to the paper of the author. Label of the each node denotes the field of the study of the corresponding author.

**Co-purchase datasets** Amazon Computers and Amazon Photo are two co-purchase networks (Shchur et al., 2018) where each node denotes products and an edge exists if two products are bought frequently. Node features denote the bag-of-words representation of the product reviews. Node labels indicate the product category.

## C  Computational Resources

We run the experiments on the datasets with the standard train/validation/test splits. The mean and standard deviations are reported after 10 runs on multiple random seeds for each dataset. All experiments are done on a single GPU GeForce RTX 3090 with 24GB memory capacity.

## D  Hyperparameter Details

In this section, we will describe the hyperparameters of every dataset employed for the experimentation. Refer Tto ables 10, 11, 12, 13, and 17 for the hyperparameters of category-wise positional encodings for MNIST, CIFAR10, PATTERN, CLUSTER, ogbg-molhiv, ogbg-ppa, ogbg-molpcba, and ogbg-code2 datasets, respectively. The hyperparameters are adjusted from the initial setting, which is inspired by SAN (Kreuzer et al., 2021), GraphGPS (Rampášek et al., 2022), SAT (Chen et al., 2022), and GraphGPS (Rampášek et al., 2022). The model parameters are optimized by the Adam (Kingma & Ba, 2014) optimizer with the default settings. The learning rate is adjusted after the number of "patience" epochs.

The hyperparameters of the Co-author and Co-purchase datasets are provided in Table 18. The dimension of the positional vector is 128 when the eigenvectors of the Laplacian matrix for every network depth initialize PEs. The dimension of PE is fixed at 8 when PEs are initialized with RWPE.

Table 10: Hyperparameters for the MNIST dataset for every category of PE generated in the experiments.

| Hyperparameters / PE Category | MNIST | | | | | | | |
|---|---|---|---|---|---|---|---|---|
| | 1 | 2 | 3 | 4 | 5 | 6 | 7 | 8 |
| # HyPE-GT Layers | | | | 4 | | | | |
| # Head | | | | 8 | | | | |
| Hidden Dim | | | | 80 | | | | |
| Curvature | | | | 1.0 | | | | |
| Activation | | | | ReLU | | | | |
| # PE Layers | | | | 2 | | | | |
| Dropout | | | | 0.0 | | | | |
| Layernorm | | | | False | | | | |
| Batchnorm | | | | True | | | | |
| PE Dim | | | | 6 | | | | |
| Graph Pooling | | | | Sum | | | | |
| Batch size | | | | 128 | | | | |
| Init LR | | | | 0.001 | | | | |
| Epochs | | | | 1000 | | | | |
| Patience | | | | 10 | | | | |
| Weight Decay | | | | 0.0 | | | | |

Table 11: Hyperparameters for CIFAR10 dataset for every category of PE generated in the experiments.

| Hyperparameters / PE Category | CIFAR10 | | | | | | | |
|---|---|---|---|---|---|---|---|---|
| | 1 | 2 | 3 | 4 | 5 | 6 | 7 | 8 |
| # HyPE-GT Layers | | | | 4 | | | | |
| # Head | | | | 8 | | | | |
| Hidden Dim | | | | 80 | | | | |
| Curvature | | | | 1.0 | | | | |
| Activation | | | | ReLU | | | | |
| # PE Layers | | | | 2 | | | | |
| Dropout | | | | 0.0 | | | | |
| Layernorm | | | | False | | | | |
| Batchnorm | | | | True | | | | |
| PE Dim | | | | 16 | | | | |
| Graph Pooling | | | | Mean | | | | |
| Batch size | | | | 128 | | | | |
| Init LR | | | | 0.001 | | | | |
| Epochs | | | | 1000 | | | | |
| Patience | | | | 10 | | | | |
| Weight Decay | | | | 0.0 | | | | |

## E  More Results on Ablation Study

We conduct ablation studies on the ogbg-molpcba and ogbg-code2 datasets from OGB. By varying different modules of HyPE-GT, we generate a diverse set of learnable hyperbolic positional encodings. Refer to Figure 11 for detailed visualization. The variation in the performances is recorded across 8 categories of PEs. The experiment underscores the utility of generating a diverse set of PEs for solving downstream tasks.

Table 12: Hyperparameters for PATTERN dataset for every category of PE generated in the experiments.

| Hyperparameters / PE Category | PATTERN | | | | | | | |
|---|---|---|---|---|---|---|---|---|
| | 1 | 2 | 3 | 4 | 5 | 6 | 7 | 8 |
| # HyPE-GT Layers | | 10 | | | | | 10 | |
| # Head | | 8 | | | | | 8 | |
| Hidden Dim | | 80 | | | | | 80 | |
| Curvature | | 1.0 | | | | | 1.0 | |
| Activation | | ReLU | | | | | ReLU | |
| # PE Layers | | 1 | | | | | 1 | |
| Dropout | | 0.0 | | | | | 0.0 | |
| Layernorm | | False | | | | | False | |
| Batchnorm | | True | | | | | True | |
| PE Dim | | 6 | | | | | 2 | |
| Graph Pooling | | Mean | | | | | Mean | |
| Batch size | | 26 | | | | | 26 | |
| Init LR | | 0.0005 | | | | | 0.0003 | |
| Epochs | | 1000 | | | | | 1000 | |
| Patience | | 10 | | | | | 10 | |
| Weight Decay | | 0.0 | | | | | 0.0 | |

Table 13: Hyperparameters for CLUSTER dataset for every category of PE generated in the experiments.

| Hyperparameters / PE Category | CLUSTER | | | | | | | |
|---|---|---|---|---|---|---|---|---|
| | 1 | 2 | 3 | 4 | 5 | 6 | 7 | 8 |
| # HyPE-GT Layers | | 10 | | | | | 10 | |
| # Head | | 8 | | | | | 8 | |
| Hidden Dim | | 80 | | | | | 80 | |
| Curvature | | 1.0 | | | | | 1.0 | |
| Activation | | ReLU | | | | | ReLU | |
| # PE Layers | | 4 | | | | | 2 | |
| Dropout | | 0.0 | | | | | 0.0 | |
| Layernorm | | False | | | | | False | |
| Batchnorm | | True | | | | | True | |
| PE Dim | | 6 | | | | | 16 | |
| Graph Pooling | | Mean | | | | | Mean | |
| Batch size | | 32 | | | | | 32 | |
| Init LR | | 0.0005 | | | | | 0.0003 | |
| Epochs | | 1000 | | | | | 1000 | |
| Patience | | 10 | | | | | 10 | |
| Weight Decay | | 0.0 | | | | | 0.0 | |

Table 14: Hyperparameters for ogbg-molhiv dataset for every category of PE generated in the experiments.

| Hyperparameters / PE Category | ogbg-molhiv | | | | | | | |
|---|---|---|---|---|---|---|---|---|
| | 1 | 2 | 3 | 4 | 5 | 6 | 7 | 8 |
| # HyPE-GT Layers | | | | 10 | | | | |
| # Head | | | | 4 | | | | |
| Hidden Dim | | | | 64 | | | | |
| Curvature | | | | 1.0 | | | | |
| Activation | | | | ReLU | | | | |
| # PE Layers | | | | 2 | | | | |
| Dropout | | | | 0.01 | | | | |
| Layernorm | | | | False | | | | |
| Batchnorm | | | | True | | | | |
| PE Dim | | | | 32 | | | | |
| Graph Pooling | | | | Max | | | | |
| Batch size | | | | 64 | | | | |
| Init LR | | | | 0.0001 | | | | |
| Epochs | | | | 1000 | | | | |
| Patience | | | | 20 | | | | |
| Weight Decay | | | | 0.0 | | | | |

Table 15: Hyperparameters for the ogbg-ppa dataset for every category of PE generated in the experiments.

| Hyperparameters / PE Category | ogbg-ppa | | | | | | | |
|---|---|---|---|---|---|---|---|---|
| | 1 | 2 | 3 | 4 | 5 | 6 | 7 | 8 |
| # HyPE-GT Layers | | | | 2 | | | | |
| # Head | | | | 2 | | | | |
| Hidden Dim | | | | 16 | | | | |
| Curvature | | | | 1.0 | | | | |
| Activation | | | | ReLU | | | | |
| # PE Layers | | | | 2 | | | | |
| Dropout | | | | 0.0 | | | | |
| Layernorm | | | | False | | | | |
| Batchnorm | | | | True | | | | |
| PE Dim | | | | 8 | | | | |
| Graph Pooling | | | | Sum | | | | |
| Batch size | | | | 16 | | | | |
| Init LR | | | | 0.0003 | | | | |
| Epochs | | | | 1000 | | | | |
| Patience | | | | 15 | | | | |
| Weight Decay | | | | 0.0 | | | | |

## F  Comparative Study on Number of Parameters

We perform a comparative study on the parameters of the existing Graph Transformers like GraphTransformer (Dwivedi & Bresson, 2020), SAN (Kreuzer et al., 2021), SAT (Chen et al., 2022), Graphormer (Ying et al., 2021), EGT (Hussain et al., 2022), and GraphGPS (Rampášek et al., 2022) with our proposed method HyPE-GT. Refer to Table **??** for detailed information regarding the number of model parameters. The number of parameters from both variants is equal because the variants are structurally identical, but they differ only in the way PEs are incorporated. For PATTERN and CLUSTER, both our variants have several parameters comparable with GraphTransformer, SAN, and EGT. But as GraphGPS is a linearized Transformer architecture. Therefore, it is desirable to have a lower number of parameters. However, the SAT has a much higher number of parameters. On the other side, our framework produces a higher number of parameters compared to EGT and GraphGPS (as the rest of the methods do not report the numbers). Still, HyPE-GT outperforms all methods. Lastly, our framework produces the lowest number of parameters compared to all SOTA approaches, and it also achieves the best performance on the dataset. As ogbg-molhiv is one of the large-scale graphs, the performance of the framework is evidence of the efficacy of the hyperbolic positional encodings.

Table 16: Hyperparameters for the ogbg-molpcba dataset for every category of PE generated in the experiments.

| Hyperparameters / PE Category | ogbg-molpcba | | | | | | | |
|---|---|---|---|---|---|---|---|---|
| | 1 | 2 | 3 | 4 | 5 | 6 | 7 | 8 |
| # HyPE-GT Layers | | | | 5 | | | | |
| # Head | | | | 4 | | | | |
| Hidden Dim | | | | 304 | | | | |
| Curvature | | | | 1.0 | | | | |
| Activation | | | | ReLU | | | | |
| # PE Layers | | | | 2 | | | | |
| Dropout | | | | 0.2 | | | | |
| Layernorm | | | | False | | | | |
| Batchnorm | | | | True | | | | |
| PE Dim | | | | 32 | | | | |
| Graph Pooling | | | | Mean | | | | |
| Batch size | | | | 512 | | | | |
| Init LR | | | | 0.0005 | | | | |
| Epochs | | | | 1000 | | | | |
| Patience | | | | 20 | | | | |
| Weight Decay | | | | 0.0 | | | | |

Table 17: Hyperparameters for the ogbg-code2 dataset for every category of PE generated in the experiments.

| Hyperparameters / PE Category | ogbg-code2 | | | | | | | |
|---|---|---|---|---|---|---|---|---|
| | 1 | 2 | 3 | 4 | 5 | 6 | 7 | 8 |
| # HyPE-GT Layers | | | | 4 | | | | |
| # Head | | | | 4 | | | | |
| Hidden Dim | | | | 16 | | | | |
| Curvature | | | | 1.0 | | | | |
| Activation | | | | ReLU | | | | |
| # PE Layers | | | | 1 | | | | |
| Dropout | | | | 0.2 | | | | |
| Layernorm | | | | False | | | | |
| Batchnorm | | | | True | | | | |
| PE Dim | | | | 8 | | | | |
| Graph Pooling | | | | Mean | | | | |
| Batch size | | | | 32 | | | | |
| Init LR | | | | 0.0001 | | | | |
| Epochs | | | | 1000 | | | | |
| Patience | | | | 20 | | | | |
| Weight Decay | | | | 0.0 | | | | |

Table 18: Hyperparameters for Co-author and Co-purchase datasets

| Hyperparameters | Amazon photo | Amazon computers | Coauthor CS | Coauthor Physics |
|---|---|---|---|---|
| Learning rate | 0.01 | 0.01 | 0.01 | 0.01 |
| PE Dim | 128/8 | 128/8 | 128/8 | 128/8 |
| Hidden Dim | 64 | 64 | 64 | 64 |
| #PE Layers | 2 | 2 | 2 | 2 |
| Activation | ReLU | ReLU | ReLU | ReLU |
| Dropout | 0.50 | 0.50 | 0.20 | 0.20 |
| Curvature | 1.0 | 1.0 | 1.0 | 1.0 |
| weight decay | 0.0005 | 0.0005 | 0.0005 | 0.0005 |
| Training Epochs | 500 | 500 | 500 | 500 |

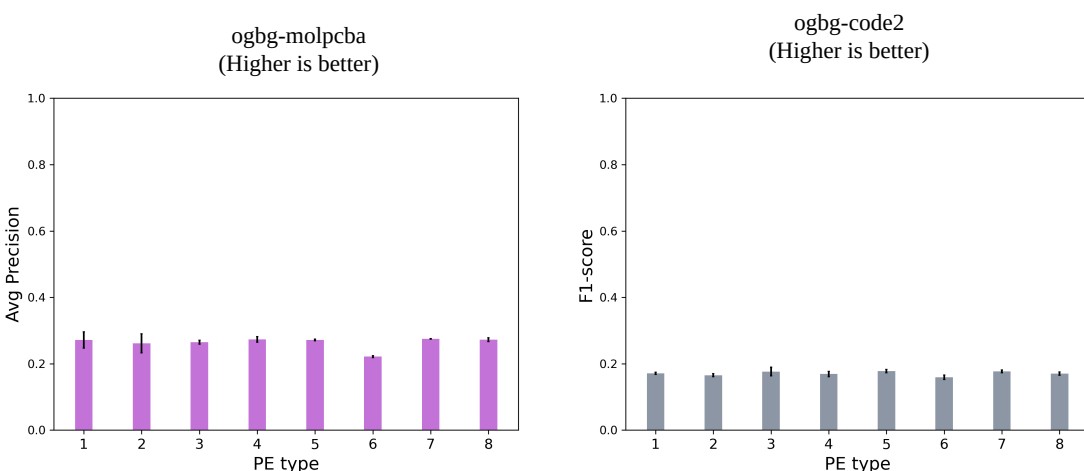

Figure 11: The performance of HyPE-GT on ogbg-molpcba and ogbg-code2 for 8 different categories of PEs is presented.

Under review as submission to TMLR

Table 19: A comparative study on the number of parameters of HyPE-GT with the other existing Graph Transformers.

| Method / Data | Init PE | Hyperbolic Manifold | Hyperbolic NN | PATTERN | CLUSTER | MNIST | CIFAR10 | ogbg-molhiv |
|---|---|---|---|---|---|---|---|---|
| GraphTransformer (Dwivedi & Bresson, 2020) | | | | 523146 | 522742 | - | - | - |
| SAN (Kreuzer et al., 2021) | | | | 507,202 | 519,186 | - | - | 528265 |
| Graphormer (Ying et al., 2021) | | | | - | - | - | - | 47.0M |
| SAT (Chen et al., 2022) | | | | 825,986 | 741,990 | - | - | - |
| EGT (Hussain et al., 2022) | | | | 500000 | 500000 | 100000 | 100000 | 110.8M |
| GraphGPS (Rampášek et al., 2022) | | | | 337201 | 502054 | 115,394 | 112,726 | 558625 |
| **HyPE-GT (ours)** | LapPE | Hyperboloid | HGCN | 524022 | 524426 | 369390 | 371150 | 389441 |
| | | | HNN | 523382 | 524426 | 369390 | 369550 | 390465 |
| | | Poincare Ball | HGCN | 523382 | 524426 | 369390 | 369550 | 388929 |
| | | | HNN | 523382 | 524666 | 369390 | 369550 | 388929 |
| | RWPE | Hyperboloid | HGCN | 523142 | 524666 | 368830 | 368990 | 390465 |
| | | | HNN | 523142 | 524666 | 368830 | 368990 | 390465 |
| | | Poincare Ball | HGCN | 523142 | 524666 | 368830 | 369790 | 390465 |
| | | | HNN | 524426 | 524666 | 368830 | 369790 | 390465 |

