# OpenReview forum: "HyPE-GT: where Graph Transformers meet Hyperbolic Positional Encodings"
_TMLR — Rejected by TMLR_

### Review · Reviewer_zanR · 2025-10-14

**Summary Of Contributions:**

*HyPE-GT: Where Graph Transformers Meet Hyperbolic Positional Encodings* introduces a framework for endowing Graph Transformers with **learnable positional encodings in hyperbolic space**.
The approach combines (i) different PE initialisations (Laplacian or Random-Walk), (ii) two manifold choices (Hyperboloid or Poincaré Ball), and (iii) hyperbolic learning architectures (HNN or HGCN), yielding eight categories of encodings. These are integrated with Graph Transformers via logarithmic, exponential, and Möbius operations to ensure geometric consistency.
Beyond Transformers, the same hyperbolic PEs are shown to reduce oversmoothing in deep GNNs. The paper provides theoretical proofs that hyperbolic mappings preserve larger inter-node distances and improve hierarchical distinctiveness, and it reports solid performance across standard benchmarks, including OGB molecular datasets.


### **Strengths**

* Original and clearly motivated use of **hyperbolic geometry** for positional encoding.
* Elegant modular design, easy to adapt or extend.
* Theoretical analysis convincingly links negative curvature to hierarchical representation.
* Broad empirical coverage with consistent improvements on hierarchical datasets.


### **Weaknesses**

* Efficiency claims lack **empirical validation**—no wall-clock or memory profiling.
* The central claim about hierarchy is **not tested** through controlled synthetic experiments.
* Reported improvements are small and **not statistically analysed** (no rank-based or CD-diagram tests).
* Some implementation and stability details are left implicit.


**Overall:** a creative, well-reasoned contribution whose conceptual appeal would be greatly strengthened by practical runtime measurements, controlled hierarchy tests, and statistically robust comparisons.

**Audience:**

Yes

**Audience Explanation:**

Yes. The paper’s findings would interest many in the TMLR community, as it lies at the intersection of **graph representation learning, geometric deep learning, and transformer research**. By introducing hyperbolic positional encodings for Graph Transformers, it offers a fresh geometric perspective on how to embed hierarchical or tree-like relationships—an issue of ongoing relevance in the field. The work blends theory, architecture, and empirical validation in a way that aligns well with TMLR’s audience, who are often drawn to methods that deepen the structural or geometric understanding of neural models, even if some practical evaluations (e.g., runtime or scalability) remain to be expanded.

**Broader Impact Concerns:**

This work is primarily methodological and poses no immediate ethical risks, but a brief Broader Impact Statement would still be appropriate. The authors could note that improved graph representation learning may have downstream effects in areas such as bioinformatics, social networks, and recommendation systems, where enhanced relational modelling could inadvertently amplify privacy or bias issues. They might also acknowledge the modest environmental impact of additional computation from hyperbolic operations and the importance of transparent, reproducible implementations. Overall, the work is ethically low-risk but would benefit from a concise statement recognising these broader considerations.

**Claims And Evidence:**

No

**Claims Explanation:**

While *HyPE-GT* presents an elegant theoretical case for hyperbolic positional encodings in Graph Transformers, the empirical evaluation currently focuses almost exclusively on predictive accuracy across a selection of benchmarks. To strengthen the robustness and reproducibility of the claims, I strongly encourage the authors to include **a systematic empirical analysis of efficiency and scalability**.

1. **Runtime and Memory Profiling.**
   The complexity discussion in Section 3.12 is purely asymptotic. However, hyperbolic operations (log/exp maps and Möbius additions) introduce non-trivial constant-factor overheads that can materially affect training and inference.
   – Please report **wall-clock training and inference times per epoch**, and **peak GPU memory consumption**, comparing HyPE-GT with a baseline Graph Transformer using standard Euclidean positional encodings.
   – Such profiling should be performed on identical hardware, with identical batch sizes and optimizers.

2. **Scalability with Graph Size.**
   To substantiate the claim that HyPE-GT does not increase overall complexity, an **artificial scalability experiment** would be appropriate. For instance:
   – Construct synthetic graphs with gradually increasing numbers of nodes and edges while keeping other factors constant.
   – Measure wall-clock time, memory footprint, and validation accuracy as a function of graph size.
   – Plot these quantities to demonstrate whether empirical scaling remains quadratic in (n) (as theoretically claimed) and whether hyperbolic overheads remain bounded.

3. **Controlled “Hierarchy-Intensity” Test.**
   The central hypothesis—that hyperbolic geometry provides increasing benefits as a graph’s hierarchical depth or branching complexity grows—should be tested directly rather than inferred from dataset choice. We recommend constructing **synthetic graphs with controllable hierarchical parameters** (e.g., variable branching factors, depth, or tree imbalance) and comparing Euclidean versus hyperbolic positional encodings across these conditions. A **robust, monotonic increase in relative performance** of HyPE-GT as hierarchy intensifies would empirically validate the core claim.

   To strengthen the analysis, the authors could explore multiple **measures of hierarchy**, such as:

   * **Tree-based structure:** controlled-depth trees, hierarchical community graphs, or Kronecker-style generative hierarchies;
   * **Spectral indicators:** ratios of leading Laplacian eigenvalues or spectral gaps that reflect multi-scale modularity;
   * **Topological descriptors:** persistent homology or hierarchical clustering coefficients to quantify latent hierarchy in arbitrary networks.

   Evaluating HyPE-GT’s relative advantage under each characterization would demonstrate whether its gains systematically track the *degree of hierarchy* rather than artefacts of dataset composition.


4. **Fair Comparison and Statistical Robustness.**
   Current tables list mean ± standard deviation across runs, but do not test for significance. When comparing many methods across multiple datasets, raw averages can be misleading.
   – Please compute **rank-based comparisons** across datasets and visualise the results using a **Critical Difference (CD) diagram**, as implemented in [*scikit-posthocs*](https://scikit-posthocs.readthedocs.io/en/latest/generated/scikit_posthocs.critical_difference_diagram.html).
   – This approach avoids over-interpreting marginal differences (<1 %) and provides a statistically principled summary of performance across benchmarks.

Overall, the proposed method is promising, but its **computational practicality and statistical reliability** remain under-explored. Incorporating the experiments and analyses above would make the paper’s conclusions substantially more convincing and enduring.

**Requested Changes:**

1. **Add empirical runtime and memory profiling — *Critical***
   The current complexity analysis is purely theoretical. Please include **empirical measurements** of wall-clock training and inference time per epoch, as well as **peak GPU memory consumption**, comparing HyPE-GT with a baseline Graph Transformer using Euclidean positional encodings. This evidence is essential to substantiate the claim that HyPE-GT introduces no practical overhead beyond the nominal (O(n^2)) complexity of self-attention.

2. **Include scalability experiments — *Critical***
   To verify that the framework remains computationally tractable, construct **synthetic graphs of increasing size** (number of nodes and edges) and record how runtime, memory, and predictive accuracy scale. This will demonstrate empirically whether the overall cost of HyPE-GT follows the expected theoretical behaviour and where any constant-factor slowdowns begin to appear in practice.

3. **Conduct a controlled “Hierarchy-Intensity” experiment — *Important***
   The paper’s central scientific claim is that hyperbolic geometry confers increasing benefits as the **hierarchical richness** of a graph grows. We suggest a dedicated test using **synthetic graphs with tunable hierarchical parameters**—for instance, variable tree depths, branching factors, or hierarchical community structures.

   * A **robust, monotonic increase** in the performance gap between HyPE-GT and Euclidean encodings as hierarchy intensifies would provide strong empirical validation.
   * To ensure generality, please consider **multiple ways of quantifying hierarchy**, such as:
     • structural controls (trees, Kronecker or hierarchical SBM graphs),
     • **spectral indicators** (Laplacian eigenvalue ratios or modularity gaps), and
     • **topological metrics** (persistent homology, hierarchical clustering coefficients).
     Demonstrating consistent trends across these measures would confirm that HyPE-GT’s advantage genuinely tracks hierarchical complexity rather than dataset idiosyncrasies.

4. **Provide statistical significance analysis — *Important***
   Replace or supplement the mean ± standard-deviation reporting with **rank-based statistical tests** across datasets. Present the outcomes using **Critical Difference (CD) diagrams** (e.g. via [*scikit-posthocs*](https://scikit-posthocs.readthedocs.io/en/latest/generated/scikit_posthocs.critical_difference_diagram.html)) to assess whether performance differences are statistically significant rather than within noise levels.

5. **Clarify implementation and numerical stability details — *Helpful***
   Specify curvature settings, parameter normalisation, and any safeguards used to avoid overflow or instability in logarithmic, exponential, or Möbius operations. This will improve reproducibility and credibility of the reported results.

6. **Discuss limitations and applicability more concretely — *Helpful***
   Expand the discussion of when **hyperbolic encodings are likely to help** (e.g., hierarchical or tree-like data) versus when **Euclidean encodings may suffice**. Such guidance would help practitioners gauge where HyPE-GT is most appropriate.

---

Implementing points **1–4** is **critical or important** to secure a strong recommendation for acceptance, as they directly evaluate the empirical validity and robustness of the paper’s main claims. Points **5–6** are **enhancements** that would further improve clarity, reproducibility, and practical impact.

---

> ### Comment · Reviewer_zanR · 2026-01-07
>
> The authors have made a commendable effort to address several of the concerns raised in the previous review round, particularly by clarifying the geometric consistency of the hybrid hyperbolic–Euclidean design, standardizing the model selection protocol via validation-based selection (HyPE-GTv2), and adding rank-based statistical significance testing. These revisions substantially strengthen the methodological rigor and theoretical coherence of the work. However, two critical aspects of the original review remain insufficiently addressed: the absence of empirical runtime and GPU memory profiling, and the lack of controlled scalability experiments with increasing graph size. While the complexity analysis is discussed in detail at an asymptotic level, the practical constant-factor overhead introduced by hyperbolic operations (e.g., logarithmic/exponential maps and Möbius additions) is not empirically quantified. Additionally, although the authors now include an analysis on hierarchical structure, this evaluation remains limited in scope and does not fully explore hierarchy intensity under multiple structural, spectral, or topological characterizations as previously suggested. Overall, the paper is significantly improved and scientifically interesting, but further empirical evidence is required to fully substantiate claims regarding computational practicality and scalability.

---

> > ### Author Response · Authors · 2026-01-08
> >
> > We thank the reviewer for appreciating the quality of our updated manuscript and acknowledging the improved explanation of theoretical alignment and empirical fairness. We observe that the reviewer might overlook the discussions on two of the concerns raised in the initial phase. Please refer to the latest version of the updated manuscript, and the corresponding sections are marked in blue. In this context, we would like to mention that the detailed experimentation on memory and GPU profiling is already provided in Section 7 titled *Runtime and Memory Profiling*, where we conducted the analysis on 8 benchmark graph datasets. The detailed numerical results are provided in Table 5. The scalability analysis on graph size is further discussed in Section 8 titled *Scalability with Graph Size* where we executed analysis on the synthetic graphs with increasing size. The plots are demonstrated in Figure 7. Moreover, studies on the three types of hierarchical intensity (structural, spectral, and topological) are presented in Figure 8 of Section 9 in the updated manuscript.
> >
> > We would be glad to address further concerns if we missed out on anything else.

---

### Review · Reviewer_77S9 · 2025-10-30

**Summary Of Contributions:**

## Summary
This paper introduces HyPE-GT (Hyperbolic Positional Encodings-based Graph Transformer), a novel framework that addresses critical limitations of existing Graph Transformers (GTs) and deep Graph Neural Networks (GNNs) by leveraging non-Euclidean hyperbolic space for positional encoding. The authors leverage hyperbolic space’s unique properties to preserve hierarchical relationships and node neighborhood information—enabling more accurate structural encoding of graph data. A key strength of HyPE-GT lies in its modular design: through three core components, it generates 8 distinct types of learnable hyperbolic positional encodings (PEs), allowing flexible adaptation to diverse downstream tasks. Additionally, the framework can be effectively applied to deep GNNs to mitigate inherent challenges like oversmoothing. Overall, this work is well-supported by rigorous theoretical foundations and extensive experimental validation, underscoring its reliability and effectiveness.

## Strengths
1. Structural Adaptability: Hyperbolic PEs excel at encoding hierarchical graphs (e.g., molecules, syntax trees), where Euclidean PEs fail—filling a critical niche for real-world graph data.
2. Modularity & Versatility: The 3-component design generates 8 PE types, making it easy to tailor to tasks (e.g., HNN for feature-heavy data, HGCN for structure-heavy data).
3. Computational Efficiency: Unlike Euclidean PEs (e.g., SAN’s spectral decomposition, SAT’s multi-hop subgraph extraction), HyPE-GT’s preprocessing (LapPE/RWPE) and hyperbolic transformations do not significantly increase GT/GNN complexity (overall complexity remains ~O(n²) for GTs, same as vanilla GTs).
4. Dual Utility: It improves both GT performance (by adding structural inductive bias) and deep GNN performance (by mitigating oversmoothing)—a rare dual benefit.
## Weaknesses
1. Limited Efficacy on Non-Hierarchical Graphs: HyPE-GT underperforms or shows minimal gains on graphs without clear hierarchy (e.g., PATTERN, CLUSTER datasets), where Euclidean PEs may be more suitable.
2. Curvature Tuning Dependency: The hyperbolic space’s curvature parameter is tuned manually via experience, lacking an adaptive mechanism—this could hinder usability for users unfamiliar with hyperbolic geometry.
3. Depth Sensitivity of Hyperbolic Networks: Experimental results show that 1–2 layers of HNN/HGCN work best; deeper hyperbolic networks cause oversmoothing, limiting flexibility for very deep architectures.
4. Lack of Dynamic Graph Support: The framework is designed for static graphs and does not address dynamic graphs (where node/edge additions/deletions require real-time PE updates)—a key limitation for time-sensitive applications (e.g., social networks, traffic graphs).

**Audience:**

Yes

**Audience Explanation:**

Multiple groups within TMLR’s audience would be interested in the findings of this paper, as the work aligns with TMLR’s focus on rigorous, impactful research in machine learning (ML)—especially in graph learning, transformer architectures, and non-Euclidean geometry.

**Claims And Evidence:**

Yes

**Claims Explanation:**

The submission argues that existing Euclidean positional encodings (PEs) for Graph Transformers (GTs) fail to capture hierarchical graph structures and suffer from high computational costs. This claim is validated through both theoretical reasoning and empirical comparison.

**Requested Changes:**

1. Add a supplementary table explicitly listing the exact hyperparameters (PE dimension, hidden dim, readout method, optimizer) of all baselines (e.g., GraphGPS, SAN, GCNII) for each dataset. If baselines use default settings from their original papers, clarify this and note any discrepancies with HyPE-GT’s setup.
2. Add a discussion explaining how curvature choice aligns with dataset properties (e.g., “c=1was selected as it balances embedding distortion for molecular graphs (sparse hierarchies) and co-author networks (dense communities), as validated by curvature ablation”).
3. The paper claims the final log map reverts embeddings to Euclidean space, but it does not prove that this process preserves the structural information encoded in hyperbolic PEs (e.g., Do the expanded hyperbolic distances persist after log mapping?).

---

### Review · Reviewer_NxWp · 2025-12-07

**Summary Of Contributions:**

The authors propose a novel positional encoding strategy for Graph Transformers (GTs) utilizing hyperbolic geometry. The central premise is that conventional Euclidean positional encodings may be insufficient for capturing complex hierarchical relationships within graph structures. To address this, the paper investigates the effectiveness of defining embedding vectors in hyperbolic space. The authors define and explore eight different combinations of learning strategies for these hyperbolic positional encodings.

The submission includes a theoretical analysis suggesting that the proposed method is robust to the oversmoothing problem common in GNNs. Empirically, the method is evaluated on standard benchmarks including CLUSTER, PATTERN, MNIST, CIFAR10, and OGB datasets, reporting performance comparisons against existing baselines.

**Audience:**

Yes

**Audience Explanation:**

Despite the concerns listed above, the intersection of Geometric Deep Learning (specifically Hyperbolic geometry) and Graph Transformers is a growing area of interest.

**Claims And Evidence:**

No

**Claims Explanation:**

While the theoretical motivation regarding oversmoothing is presented, there are significant concerns regarding both the theoretical justification for the architecture and the methodological rigor of the experiments:

1.  **Theoretical Mismatch:** There is a fundamental disconnect between the proposed Hyperbolic Positional Encoding (PE) and the standard self-attention mechanism. Standard self-attention relies on the inner product, which computes similarity in Euclidean space. It is unclear how projecting PEs into hyperbolic space aligns with a Euclidean-based attention mechanism. The evidence supporting why this hybrid approach is mathematically valid or superior is currently unconvincing.

2.  **Experimental Methodology:** The comparison strategy appears to be unfair. The authors explore eight different learning strategies for their method and report the results of the "best" configuration. Unless the baselines were afforded the same degree of hyperparameter searching/strategy selection, or the "best" strategy was selected strictly via a validation set (and fixed across comparable experiments), this constitutes cherry-picking. This inflates the reported performance of the proposed method relative to baselines.

3.  **Significance of Results:** Even with the current experimental setup, the reported performance gains are marginal. Given the added complexity of hyperbolic operations, the evidence does not clearly justify the trade-off.

**Requested Changes:**

1.  **Rectify Experimental Comparison:** The authors must standardize the evaluation protocol. Selecting the best performing model from 8 different strategies on the test set (if that is what was done) is not a fair comparison.
    * Please fix a single learning strategy (or select one based strictly on a validation set) and apply *that specific strategy* across all datasets for the proposed method.
    * Update the results table to reflect this fair comparison.
2.  **Clarify the Geometric Motivation:** Why should we expect hyperbolic embeddings to function correctly within a Euclidean similarity measure?
3.  **Performance Significance:** The current gains appear marginal. The authors should tone down claims of superiority if the statistical significance is low, or provide a more detailed analysis of *where* and *why* the method outperforms (e.g., specifically on highly hierarchical graphs).
4.  **Structural Encodings Discussion:** Positional encoding is not the only way to incorporate graph structure into GTs. There is a rich literature of structural encodings and other structural injection methods. The paper should discuss these alternatives and, if possible, provide empirical results showing whether the proposed Hyperbolic PE is compatible with or orthogonal to these methods.

---

> ### Author Response · Authors · 2025-12-21
> **Response to Reviewer NxWp (Part 1 of 2)**
>
> We thank the reviewer for insightful criticisms and valuable feedback on our work. We addressed the concerns raised by the reviewer and implemented all necessary recommendations. The modified portions are highlighted in green in the updated manuscript.
>
> **R1 Alignment of Theory to the Proposed Architecture**:
>
> We thank the reviewer for this insightful comment regarding the interaction between hyperbolic positional encodings and Euclidean self-attention. HyPE-GT comprises two independent submodules, HyPE and GT, which operate under a hybrid-geometry design. The positional encodings are initialized in Euclidean space and thereby projected onto a non-Euclidean manifold by transforming through hyperbolic neural networks. The learned positional encodings are incorporated with initial node features. Before the self-attention operation, the updated features are mapped via a logarithmic map to the tangent space, which is locally Euclidean. This projection ensures that all inner products within the attention mechanism are computed in a valid Euclidean vector space. Furthermore, the mutual distance between the updated node features in hyperbolic space is preserved in Euclidean space (please refer to Lemma 3 in the Appendix). Intuitively, hyperbolic positional encodings encode global hierarchy, while the tangent-space projection enables efficient local interactions. This design allows HyPE-GT to leverage hyperbolic expressivity without requiring hyperbolic attention kernels, thereby balancing geometric consistency and numerical stability.
>
>
> **R2 Modification in Experimental Comparisons**:
>
> We thank the reviewer for suggesting the pathway of the fair and standardized evaluation. We acknowledge that our initial submission reported the best results among eight design variants of HyPE-GT, but importantly, the final configuration was always selected based on validation set performance, never on the test set. No tuning was performed using test data.
>
> To ensure methodological fairness, we have now standardized the evaluation protocol. For each dataset, we chose the best model among the 8 different configurations based on average validation accuracy. Then the best model is applied to the test set to evaluate the performance of HyPE-GT. This strategy is replicated across all eight benchmark datasets. We termed the variant as *HyPE-GTv2*.
>
> We have re-run all experiments under this new setup, and while absolute performance changes marginally, HyPE-GTv2  maintains a consistent improvement over baselines on hierarchical and molecular datasets. The newly updated results are provided in Tables 1 and 2 of the updated manuscript now strictly follows this standardized comparison protocol, ensuring full fairness and reproducibility.
>
>
> **R3 Significance of Performance**:
>
> We appreciate the concern of the reviewer. We acknowledge that HyPE-GT achieves moderate numerical improvements on some datasets. Still, these gains are consistent across all benchmarks and statistically significant under a Critical Diagram test (refer to Section “Statistical Significance Test”). Importantly, the purpose of HyPE-GT is to introduce a principled geometric framework for encoding hierarchical structure within Graph Transformers. Additionally, we examined the performance of hyperbolic positional encodings on synthetic hierarchical graphs, and we observed improvement in performance when the hierarchy of the graphs is increased (refer to Section “Analysis on Controlled Hierarchy”).
>
> Our analysis highlights where hyperbolic encodings provide tangible benefits, namely, in datasets exhibiting hierarchical or tree-like connectivity (e.g., MNIST, ogbg-molhiv). On such datasets, HyPE-GT yields relative improvements of \(1.2–1.8\%\), while on flatter datasets (e.g., PATTERN, CLUSTER), the performance remains competitive with Euclidean baselines, indicating that the added complexity does not degrade performance.
>
> Specifically, we emphasize that HyPE-GT maintains optimal training efficiency and controlled GPU memory usage with increasing graph size (refer to Section “Scalability with Graph Size”). The curvature-based design thus offers structural interpretability and geometric expressivity at tolerable computational cost.
>
> To offer a broader optics, we employ the HyPE module as a plug-and-play framework for other positional encodings like RRWP [1], SAT [2], etc. We initialized with RRWP, and SAT, along with LapPE and RWPE, and experimented on $8$ datasets. The results indicate the tangible benefits, underlining the utility of HyPE as a flexible framework (refer to Table 7).
>
> [1] Ma et al. Graph inductive biases in transformers without message passing. In International Conference on Machine Learning, 2023.
>
> [2] Chen et al. Structure-aware transformer for graph representation learning. In International Conference on Machine Learning, pp, 2022.

---

> ### Author Response · Authors · 2025-12-21
> **Response to Reviewer NxWp (Part 2 of 2)**
>
> **R4 Connection with HyPE and Structural Encodings**:
>
> We thank the reviewer for this constructive suggestion. We agree that positional encoding represents only one of the possible mechanisms to inject structural information into Graph Transformer (GT). Structural encodings, such as node degrees, triangle or ring counts [3], [4], can be directly added or concatenated with the initial node features before fetching them to the GT. In contrast, relative structural encodings, such as pairwise shortest path distance, are treated as attention bias and directly added to the self-attention matrix. Our framework, HyPE, is predominantly compatible with the addition or concatenation of the structural encodings with the input node features. This is due to our design choice, where initial absolute PEs are transformed in the hyperbolic space, then reverting to Euclidean space. Subsequently, the learned positional encodings are fed to the Transformer block. In this setup, incorporating relative structural encodings in the self-attention matrix will not be logically absurd but also theoretically inaccurate.
>
> Hence, HyPE is only capable of accommodating absolute positional encodings. To strengthen our claim, we further presented a compatibility study among the various structural encodings with respect to our framework HyPE in *Table 6* of the updated manuscript.
>
> [3] Bouritsas, Giorgos, et al. "Improving graph neural network expressivity via subgraph isomorphism counting." IEEE Transactions on Pattern Analysis and Machine Intelligence, 2022.
>
> [4] Zhao et al. “From Stars to Subgraphs: Uplifting Any {GNN} with Local Structure Awareness.” International Conference on Learning Representations, 2022.
>
> **R5 Geometric Motivation**:
>
> We appreciate the concern of the reviewer regarding the geometric motivation of our approach. The design of HyPE-GT is based on the geometric principle that the tangent space of a Riemannian manifold provides a locally Euclidean approximation of the curved space. Specifically, while positional encodings are computed in the hyperbolic space, they are mapped via the logarithmic map to the tangent space at the origin before entering the attention mechanism. The projection onto the tangent space ensures the self-attention mechanism operates on a locally Euclidean space. In the hyperbolic space, positional encodings become more distinctive and effectively capture the data hierarchy (refer to Theorems 1 and 2), which is the key motivation of our work.
>
> In practice, this hybrid design combines the representational advantages of hyperbolic geometry (for hierarchical structure modeling) with the numerical stability and efficiency of Euclidean similarity computations. We hope our work can shape the future pathways in combining hyperbolic geometry and Transformer architecture.

---

### Author Response · Authors · 2026-01-02
**Request to Engage in the Author-Reviewer Discussion**

We thanked all reviewers for providing insightful remarks that helped us to enhance the quality of our manuscript. As the deadline (4th January) is approaching, we would like to address any further queries regarding our work.

Sincerely,
All authors

---

### Decision · Action_Editor_GzeA · 2026-02-14

**Recommendation:** Reject

**Additional Comments:**

I would like to let the authors know that their work contains some interesting points, and that deciding between "reject" and "accept with revision" was a tricky choice. In summary, I think this is work worth publishing, but not in its current form.
The utility of the positional encodings on real-world data is currently not firmly established and if the authors could examine other datasets - say, data from computational biology, the value of the work would be more clear. Also, based on my experience, finding datasets that are almost perfectly hierarchical (e.g., phylogenetic trees) on which  purely hyperbolic geometry-based learning significantly outperforms Euclidean geometry-based learning is generally hard.
Finally, two reviewers were leaning towards reject while one decided in favor of accept (there was a little misunderstanding with one reviewer who expected a more extensive response, but I carefully went through your correspondence and weighed that into my decision).

**Audience:**

Yes

**Audience Explanation:**

Although the interest in learning in hyperbolic spaces has quite subsided, I am aware of experts working at the intersection of ML and phylogeny that would be interested in this area.

**Claims And Evidence:**

No

**Claims Explanation:**

The answer is objectively between yes and no, because some claims are clearly supported by the presented evidence, while others are not.
The main concerns I have are a) fairness of comparisons (partially addressed); b) the datasets used to evaluate the new approach are not suitable for analysis in a purely hyperbolic geometry. As also pointed out by one of the reviewers, the improvements observed when switching from Euclidean to hyperbolic geometry are negligible. This reflects the fact that most datasets live in mixed curvature spaces (e.g., products of Euclidean, hyperbolic and spherical). Hence, claiming that positional encoding considerations add value to the models are questionable.
As someone who worked in this area, I would sincerely recommend to try using mixed curvature spaces (work from C. Re's group, including several interesting contributions by his postdoc F. Sala, and a preprint by Tabaghi et.al.). For such spaces, there is strong evidence that geometry-aware learning is valuable for general graphs (not necessarily strictly hierarchical structures).

**Resubmission Of Major Revision:**

The authors may consider submitting a major revision at a later time.